# *Anopheles* homing suppression drive candidates exhibit unexpected performance differences in simulations with spatial structure

**Samuel E Champer[1†], Isabel K Kim[1†], Andrew G Clark[1,2], Philipp W Messer[1], Jackson Champer[3]***

[1]Department of Computational Biology, Cornell University, Ithaca, United States; [2]Department of Molecular Biology and Genetics, Cornell University, Ithaca, United States; [3]Center for Bioinformatics, School of Life Sciences, Peking-Tsinghua Center for Life Sciences, Peking University, Beijing, China

**Abstract** Recent experiments have produced several *Anopheles gambiae* homing gene drives that disrupt female fertility genes, thereby eventually inducing population collapse. Such drives may be highly effective tools to combat malaria. One such homing drive, based on the *zpg* promoter driving CRISPR/Cas9, was able to eliminate a cage population of mosquitoes. A second version, purportedly improved upon the first by incorporating an X-shredder element (which biases inheritance towards male offspring), was similarly successful. Here, we analyze experimental data from each of these gene drives to extract their characteristics and performance parameters and compare these to previous interpretations of their experimental performance. We assess each suppression drive within an individual-based simulation framework that models mosquito population dynamics in continuous space. We find that the combined homing/X-shredder drive is actually less effective at population suppression within the context of our mosquito population model. In particular, the combined drive often fails to completely suppress the population, instead resulting in an unstable equilibrium between drive and wild-type alleles. By contrast, otherwise similar drives based on the *nos* promoter may prove to be more promising candidates for future development than originally thought.

**\*For correspondence:**
jchamper@pku.edu.cn

†These authors contributed equally to this work

## Editor's evaluation

This is a thorough, fundamental study assessing suppression gene drives against mosquitos. The models specifically consider the spatial dynamics of gene drives and whether a form of group selection may prevent the drive from eradicating the population, with mosquito ecology parameters, leading to compelling results. This manuscript will be of interest to those working in the technical development of gene drives, those predicting how such genetically modified insects would spread in the wild, and those evaluating the technology from regulatory and funding standpoints.

## Introduction

Malaria reduction strategies based on gene drives (*Wang et al., 2022*; *Hay et al., 2021*; *Quinn and Nolan, 2020*; *Dhole et al., 2020*) in *Anopheles* mosquitoes have made substantial advances (*Simoni et al., 2020*; *Kyrou et al., 2018*; *Adolfi et al., 2020*; *Carballar-Lejarazú et al., 2020*), with several population suppression drives targeting female fertility genes recently proving successful in laboratory

settings (*Simoni et al., 2020*; *Kyrou et al., 2018*). This raises the possibility that such drives may soon be considered for field deployment. There is thus considerable incentive to engineer the optimal drive for a maximally successful test, which could potentially lead to a more wide-scale field deployment. Because malaria kills over 400,000 people every year while infecting over 200 million (*World Health Organization, 2019*), even a modest increase in the efficiency of a gene drive could correspond to a substantial decrease in new cases per year.

Despite considerable progress in the field, we still lack an in-depth understanding of how various drive and population characteristics could affect the outcome of a drive release into a natural population. Initial modeling studies assuming a panmictic population indicated that if a homing suppression drive targeting a female fertility gene can avoid the development of resistance alleles that preserve the function of the gene, it can eliminate or at least reduce the population. The exact level of suppression is a function of both species-specific ecological factors and the suppressive power of the drive. Suppressive power is often characterized in terms of the drive's 'genetic load', which for *Anopheles* female-fertility homing drives is typically defined as the fractional reduction in average fertility of a population in which the drive has reached its equilibrium frequency as compared to an otherwise identical wild-type population. In general, if the genetic load is high enough, panmictic models predict complete population elimination, and any increase in genetic load beyond that threshold can at most provide a decrease in the time to elimination (*Deredec et al., 2011*). This suggests that the differences between existing drive candidates that meet this threshold should be minimal.

However, spatially explicit models have indicated that outcomes of a suppression drive release can be substantially more complicated than those predicted by panmictic population models. In particular, it has been shown that population structure can substantially delay or prevent complete population suppression even by drives with a sufficient genetic load to reliably induce population collapse in a panmictic population (*Champer et al., 2021a*; *North et al., 2020*; *North et al., 2019*; *Bull et al., 2019*; *Champer et al., 2021b*). One mechanism that can prevent population collapse is 'chasing', a phenomenon where wild-type individuals recolonize regions where the drive has eliminated the population. In this situation, the recolonizing individuals can benefit from the reduced competition due to the low population density in these areas and can substantially increase in number before the drive once again spreads into the area and causes local suppression to reoccur. In this manner, drive and wild-type alleles can persist indefinitely, following an irregular pattern of local suppression and recolonization. This chasing phenomenon seems to be a common feature of spatial models for many types of suppression drive, regardless of whether the model is implemented using abstract spatial patches (*Bull et al., 2019*), networks of linked demes (*North et al., 2020*; *North et al., 2019*; *Birand et al., 2022*), or continuous space (*Champer et al., 2021a*; *Champer et al., 2021b*; *Faber et al., 2021*; *Liu and Champer, 2022*; *Liu et al., 2022*; *Paril and Phillips, 2022*). Unlike panmictic models, these spatial models predict that even modest differences in efficiency between drives can potentially have large effects on the outcome of a drive release, meriting careful consideration of drive candidates to identify those with the greatest potential for success in realistic conditions.

Thus far, several candidates for female-fertility homing suppression drives have been tested in *Anopheles gambiae*. Early drives with the *vasa* promoter offered high germline drive conversion efficiency, but they are not viable candidates due to high levels of somatic Cas9 activity and high rates of embryo resistance allele formation from maternally deposited Cas9 and gRNAs (*Hammond et al., 2017*; *Hammond et al., 2016*). Use of the *zpg* and *nos* promoters was shown to greatly reduce both embryo resistance allele formation and female fitness costs from somatic activity (*Hammond et al., 2021*). A drive that combined the *zpg* promoter for Cas9 with a highly conserved gRNA target site in the *dsx* gene (thus preventing the formation of functional resistance alleles) was able to successfully suppress a cage population of mosquitoes (*Kyrou et al., 2018*). A follow-up study (*Simoni et al., 2020*) included a previously developed X-shredder in this drive (*Galizi et al., 2014*) to create a male-biased population and was similarly successful in a cage study. Overall, these studies brought forward multiple potential drive candidates that have either already succeeded at suppressing cage populations or which could be expected to do so were they to be implemented with *dsx* as a target.

Here, we assess which of these drive candidates may be most successful at suppressing natural populations of *A. gambiae*. We build upon previous work, obtaining multiple possible parameterizations of these drives based on a new analysis of available experimental data. We then analyze the

**Table 1.** Drive characteristics.

| *zpg* promoter | *nos* promoter |
|---|---|
| Shared characteristics: | Shared characteristics: |
| Female HDR cut rate = 0.99 | Female HDR cut rate = 0.99 |
| Male HDR cut rate = 0.96 | Male HDR cut rate = 0.98 |
| Female germline resistance rate = 0.01 | Female germline resistance rate = 0.01 |
| Male germline resistance rate = 0.02 | Male germline resistance rate = 0.01 |
| Maternal embryo resistance rate = 0.08 | Maternal embryo resistance rate = 0.14 |
| **zpg drive** | **nos drive** |
| Paternal Cas9 deposition rate = 0.69 | Female somatic fitness cost = 0.45 |
| Female somatic fitness cost = 0.3 | Male somatic fitness cost = 0.45 |
| **zpg2 drive** | **nosF drive** |
| Female somatic fitness cost = 0.5 | Female somatic fitness cost = 0.45 |
| **zpgX drive** | |
| Paternal Cas9 deposition rate = 0.69 | |
| Female somatic fitness cost = 0.3 | **nosF2 drive** |
| X-shredding rate = 0.93 | Female somatic fitness cost = 0.15 |
| **zpg2X drive** | **nosF3 drive** |
| Female somatic fitness cost = 0.5 | No somatic fitness costs. |
| X-shredding rate = 0.93 | |

drives in the context of our previously established individual-based simulation model with continuous space (*Champer et al., 2021a*), as well as a new *Anopheles*-specific model with weekly time steps.

## Results

### Parameterization of *Anopheles* suppression gene drives

Several suppression drives have been constructed by the Crisanti lab in *Anopheles gambiae* (*Simoni et al., 2020*; *Kyrou et al., 2018*; *Hammond et al., 2016*; *Hammond et al., 2021*). By examining experimental data collected in these studies, particularly drive inheritance and viable larvae per female, we calculated drive conversion rates (the rate at which drive alleles are converted to wild-type alleles in the germline) in female and male drive/wild-type heterozygotes, and we similarly obtained estimates of the germline resistance allele formation rates (distinctly in females and males), embryo resistance allele rates from parental deposition of Cas9 and gRNAs (both maternal and paternal), and additional fitness costs (see Supplemental Results for details). Note that these data is subject to various levels of uncertainty due to data type and sample size. We assumed that fitness costs were from somatic expression of Cas9 and gRNA in drive/wild-type heterozygotes, with no intrinsic fitness cost of the drive itself (such costs appear to be small based on previous studies *Adolfi et al., 2020*; *Carballar-Lejarazú et al., 2020*; *Champer et al., 2020c*; *Champer et al., 2020a*; *Oberhofer et al., 2019*). *Table 1* contains parameterizations of each modeled drive, and *Figure 1* demonstrates how each parameter and drive mechanism operates.

The first drive to demonstrate suppression of a cage population targeted a conserved site of *dsx* using Cas9 expressed by the *zpg* promoter (*Kyrou et al., 2018*), which was also assessed in a drive with another target site (*Hammond et al., 2021*). We model the authors' interpretation of this study with paternal Cas9 deposition (zpg) and another interpretation that instead assumes no paternal deposition and higher somatic fitness costs (zpg2), which we consider more likely (Supplemental Results).

A follow-up study by the same group added the I-PpoI nuclease to the drive, thus causing it to shred the X-chromosome and bias the population toward males (*Simoni et al., 2020*). According to their data, 93% of X-chromosomes are effectively shredded in the germline. We model this variant of the drive with both the original and alternate parameter sets for the *zpg* suppression drive (zpgX and zpg2X, respectively).

The *nos* promoter has also been shown to support highly efficient homing suppression drives, though it has not yet been tested at *dsx*. We parameterize this drive (nos) based on a previous study (*Hammond et al., 2021*) with three additional alternative interpretations of the data (Supplemental Results). In one (nosF), we assume no effect of somatic Cas9 expression in males, which we believe

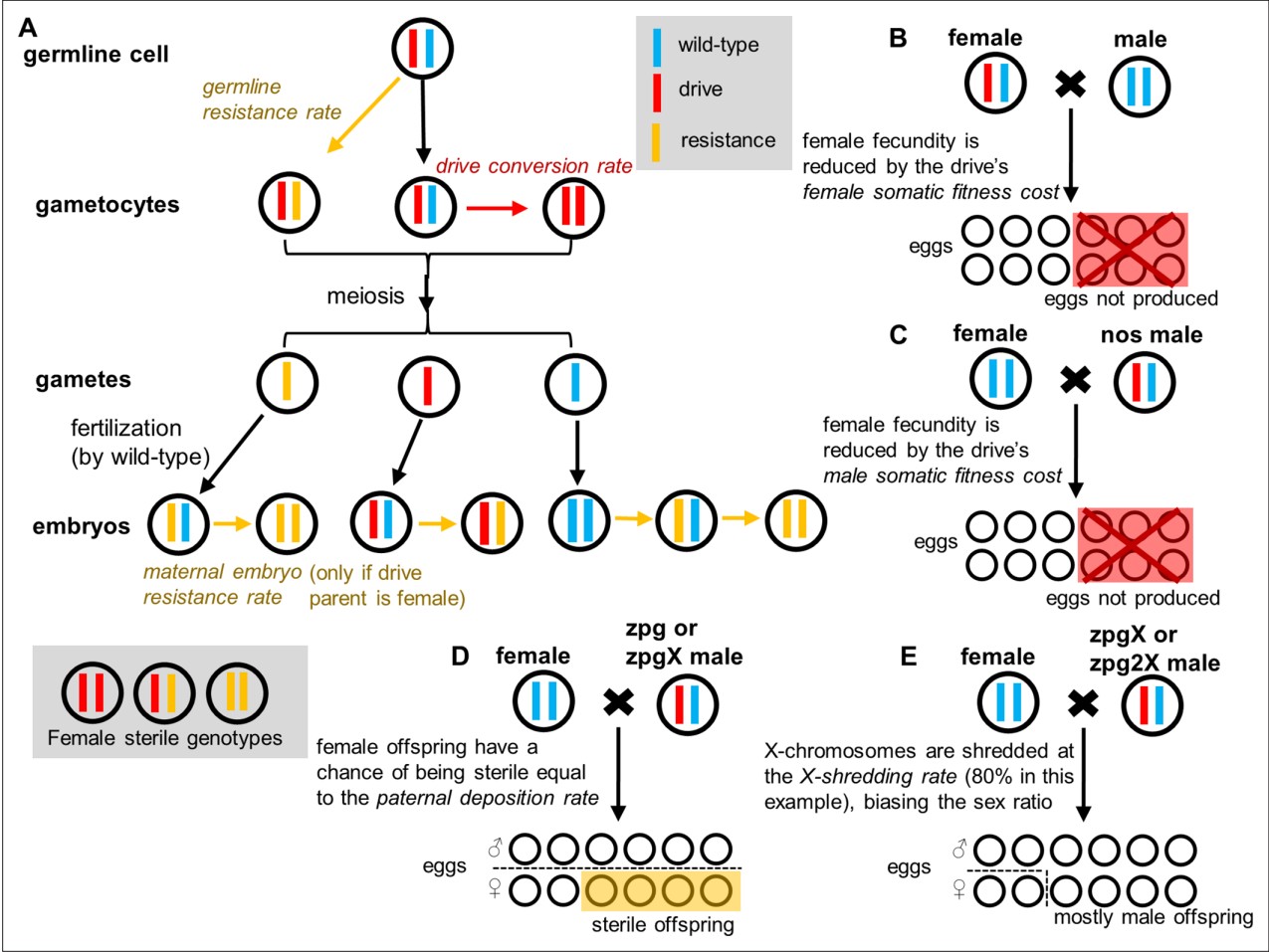

**Figure 1.** Drive mechanisms and effect of drive parameters. (**A**) Germline resistance allele formation occurs first in drive heterozygotes during reproduction. Remaining alleles can undergo drive conversion. If the mother has a drive allele, wild-type alleles in the offspring can be converted to resistance alleles in the early embryo, regardless of whether the offspring inherited a drive allele. Because all resistance alleles are assumed to be nonfunctional, any female genotype lacking at least one wild-type allele is sterile. (**B**) The fecundity of female drive heterozygotes is directly reduced by female somatic fitness costs. (**C**) The fecundity of any female is reduced if she mates with a nos male heterozygote. (**D**) Female progeny from male zpg or zpgX carriers may be sterile if paternal deposition occurs. (**E**) If the male parent has the zpgX or zpg2X drive, then X-shredding will result in an increased fraction of male progeny.

may be more realistic. A second possible alternative interpretation assumes reduced somatic effects in females (nosF2), and a third highly optimistic interpretation assumes no somatic effects at all (nosF3).

## Drive performance in the panmictic discrete-generation model

We first simulated the drives in our panmictic discrete-generation model to assess their basic properties, starting with genetic load. Genetic load describes the reduction in reproductive capacity of the population compared to a population that is identical except for being composed entirely of wild-type individuals. In panmictic populations, this measurement often reaches an early peak as the drive allele reaches its maximum frequency, but then slightly declines to a steady value once the drive allele and nonfunctional resistance alleles reach their equilibrium frequency. The rate at which wild-type alleles are converted to drive alleles in the germline of drive heterozygotes is a primary determinant of genetic load. Negative fitness effects associated with the drive can reduce the genetic load, as can the rate at which nonfunctional resistance alleles are formed in both the germline and embryo, though the effect of such alleles is usually not large (*Beaghton et al., 2019*). To eliminate a panmictic population, the drive must induce a sufficiently high genetic load in order to overpower the increased population growth rate at low density. All of the implementations of the *nos* drive were found to have a higher genetic load than the *zpg* drives (*Figure 2*), although zpg2 performed well compared to the other

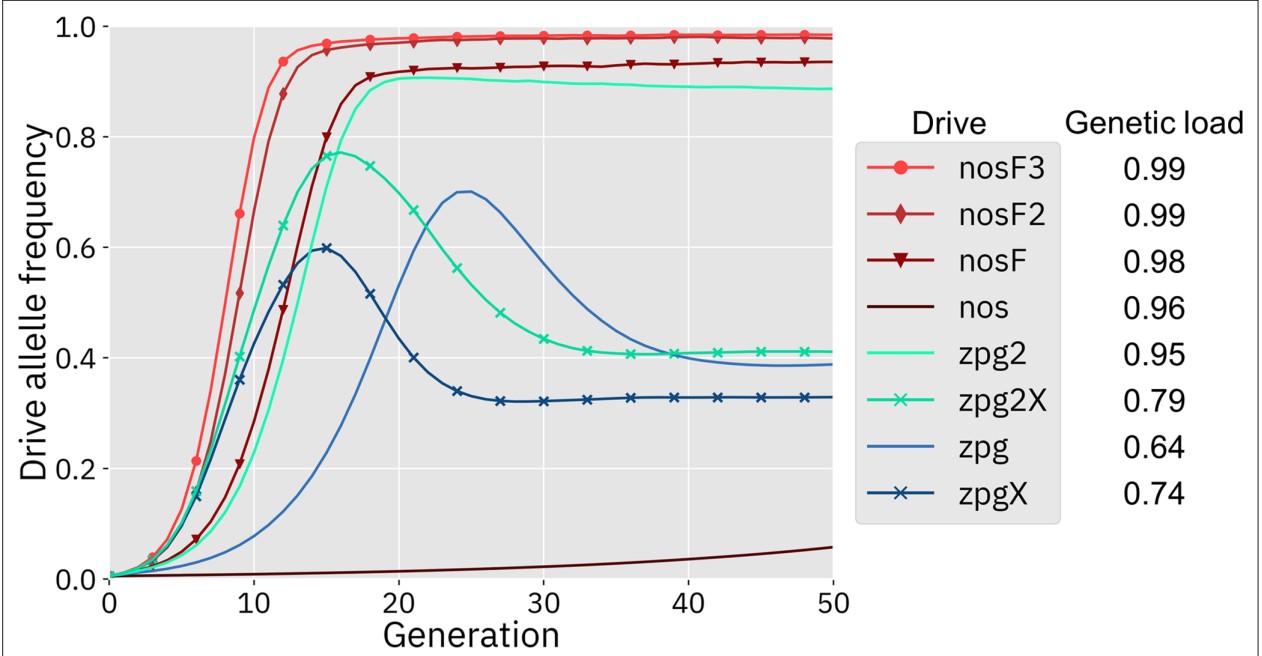

**Figure 2.** Drive allele frequency trajectories in the panmictic discrete-generation model. Using default parameters, each drive was released into a panmictic population at 0.5% initial frequency (1% heterozygote release). The average allele frequency as estimated over 100 replicates per drive is plotted per generation. Offspring were artificially generated from fertile individuals at high rates to prevent complete population suppression even at high drive frequencies and genetic loads. For a description of this method for measuring genetic load, see the Supplemental Methods.

*zpg* drives. Notably, the addition of an X-shredder was detrimental to zpg2, with zpg2X's genetic load reduced by 0.16. Even the highest fitness cost interpretation of *nos* showed a genetic load of 0.96, with the lower fitness cost interpretations scoring even higher. Three of the drives with low equilibrium genetic loads (zpg, zpgX, and zpg2X) actually had higher peak genetic loads shortly after their release, with the genetic load eventually declining to a lower equilibrium due to increased formation of nonfunctional resistance alleles.

Next, we measured the rate at which the drive spread through the population (*Figure 2*). The addition of the X-shredder substantially improved the speed of both *zpg* drives, with the zpg2 interpretation resulting in considerably higher performance than zpg. This is because the X-shredder was mostly present in males, allowing it to avoid somatic fitness costs in females. The nos drive, with its high fitness cost in both males and females, performed far worse than all the other drives in this regard, seemingly belying its high genetic load; the somatic fitness costs had so great an impact on fertility that the drive could make little headway given a low starting frequency (though the drive still eventually reaches a high equilibrium frequency, see *Appendix 1—figure 1*). However, the other *nos* drives performed well, with even nosF outperforming zpg2, although both *zpg* drives with X-shredders spread faster initially.

## Spatial discrete-generation model

Previously, we found that drive outcomes in a model with continuous space can substantially differ from those in panmictic populations (*Champer et al., 2021a*). In our spatial discrete-generation model, the migration value controls the radius in which a female can find a mate as well as the displacement between a mother and her offspring. The low-density growth rate is a multiplier on female fecundity in the absence of competition. To examine how each drive would behave under various ecological assumptions, we varied these two parameters and recorded whether the simulations resulted in (a) complete suppression without any period of chasing, (b) suppression after a period of chasing, (c) long-term chasing (defined as ongoing chasing when the simulation ends after 1000 generations), (d) drive loss after a period of chasing, or (e) drive loss without chasing (in both drive loss outcomes, the population would be quickly restored to its equilibrium after several generations). In all situations in which chasing occurred, the population size was reduced, though the magnitude of this reduction

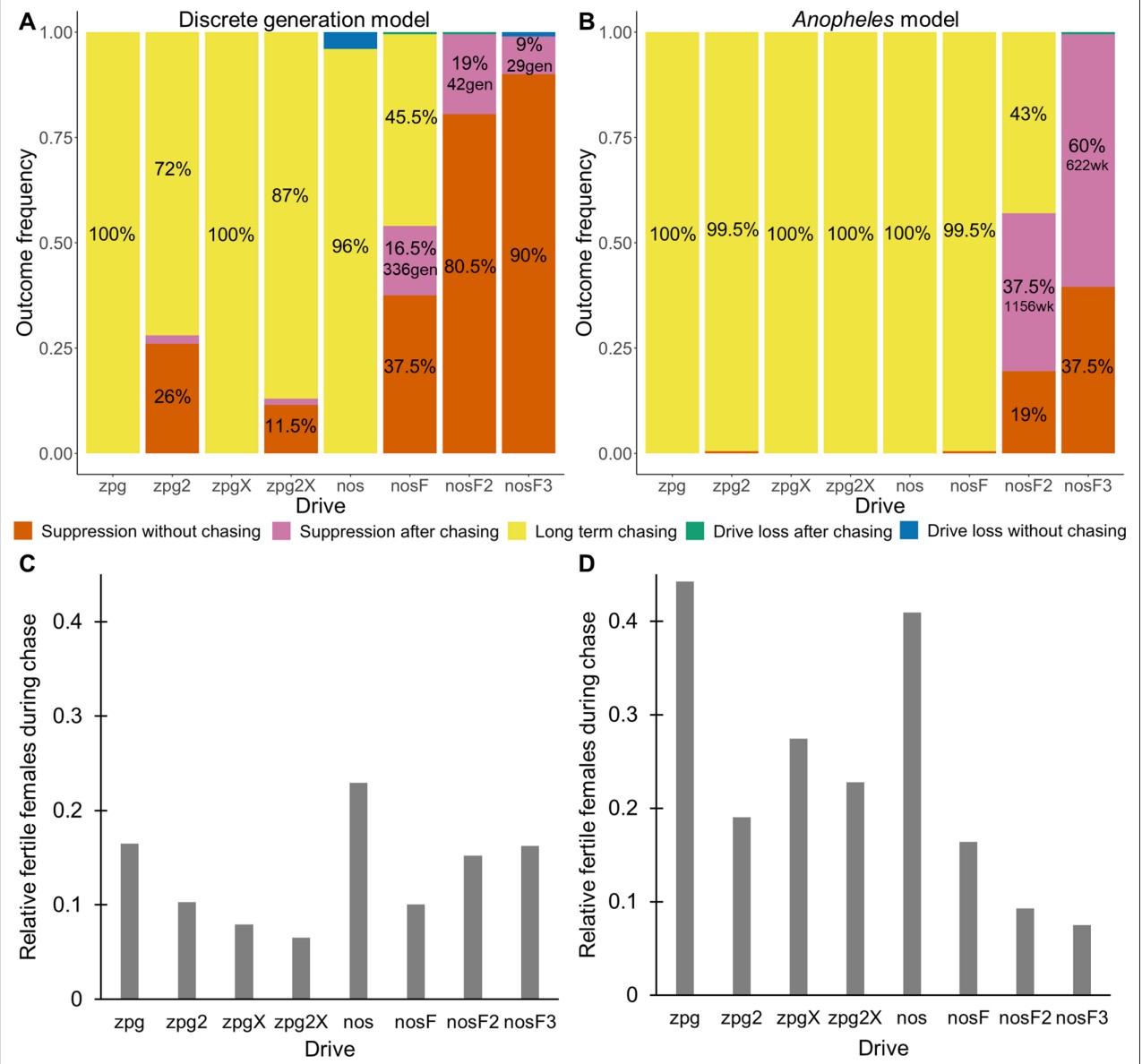

**Figure 3.** Outcomes in the spatial models. Using default parameters, a low-density growth rate of 10, and with 200 replicates per drive, each drive was released into the middle of a wild-type population. The outcome was recorded after 1000 generations or when the population was eliminated for the discrete-generation (**A**) and *Anopheles*-specific (**B**) models. In outcomes involving chasing followed by suppression, the number of generations (gen) or weeks (wk) between the start of chasing and population elimination is shown. Also displayed is the relative number of fertile females during periods of chasing (including both long-term and short-term chases) compared to the starting amount prior to release of the drive for the discrete-generation (**C**) and *Anopheles*-specific (**D**) models. Due to the high number of replicates, the error for each data point is negligible, except for the nosF2 and nosF3 drives in the discrete-generation model due to the short duration of chasing.

varied significantly from drive to drive. To this end, we also report the relative average number of fertile females (based on their genotype) starting from the beginning of the period of chasing for each parameter set, with each replicate weighted by the duration of the chase. By 'relative', we refer to the absolute number of fertile females during chasing compared to the absolute number of wild-type females present before release of the drive (the number of sterile females plays no role in this consideration). As in a previous study on chasing (*Champer et al., 2021a*), the release of certain drives can cause as much as an order of magnitude decrease to this number.

As observed in our earlier study on suppression gene drives in continuous space (*Champer et al., 2021a*), the low-density growth rate had a large impact on drive performance. *Figure 3* and

*Appendix 1—figure 2* display the result of 200 simulations at realistic parameter values for each drive (with a low-density growth rate of 10 and 6, respectively). Under both parametrizations, the zpg drive was unable to fully eliminate the population; instead, long-term chasing occurred in 100% of simulations. However, our alternative parameterization of the drive, zpg2, achieved suppression 28% of the time when the low-density growth rate was 10 (*Figure 3A*) and 63% of the time when the low-density growth rate was reduced to 6 (*Appendix 1—figure 2A*). Similar to zpg, the zpgX drive resulted in long-term chasing in almost all simulations, though the average number of fertile females was substantially reduced. However, the alternative parametrization, zpg2X, was less likely to result in long-term chasing, and suppression was observed in 12% of outcomes when the low-density growth rate was 10 (*Figure 3A*) and 23% of outcomes when it was reduced to 6 (*Appendix 1—figure 2A*).

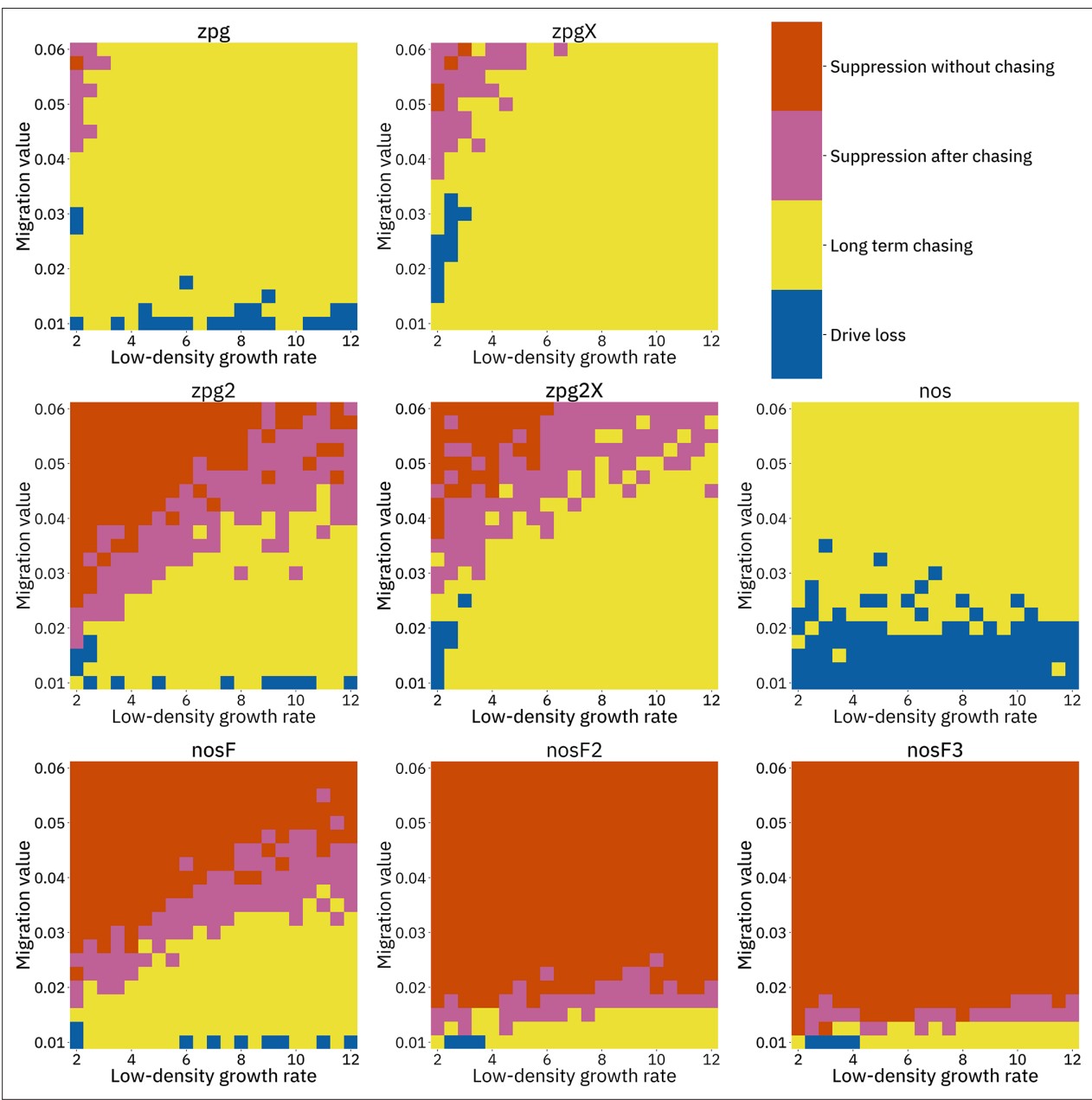

**Figure 4.** Impact of low-density growth rate and migration on outcomes in the discrete-generation model. The color of each square represents the outcome from among 20 simulations, after adjustment to show the most representative outcome. The adjustment counts pairs of 'suppression without chasing' and 'long-term chasing' outcomes as two instances of 'suppression after chasing'.

This is a substantially lower rate than the zpg2 drive, but the average number of fertile females during chases was also somewhat lower.

The standard nos drive resulted in long-term chasing in almost 100% of simulations with both low-density growth rate values (*Figure 3A*, *Appendix 1—figure 2A*). However, when the somatic fitness cost in heterozygote males was omitted (nosF), drive performance dramatically improved. When the low-density growth rate was 10, nosF achieved suppression in 54% of simulations (*Figure 3A*), and when it was lowered to 6, the suppression rate increased to 89%, a notable improvement over the best performing *zpg* drive. With a decreased female heterozygote somatic fitness cost in the nosF2 and nosF3 drive, suppression was almost guaranteed.

To further examine the behavior of these drives, we varied the migration rate from 0.01 to 0.06 and the low-density growth rate from 2 to 12. The results were broadly consistent with the results of our earlier study (*Champer et al., 2021a*). *Figure 4* shows the most common outcome from the simulations, and *Appendix 1—figures 3–6* show the likelihood of each outcome. In general, suppression tended to be increasingly likely at higher migration values and when low-density growth rates were smaller (*Appendix 1—figure 3*). There was usually a 'transition' regime (*Figure 3A*) involving suppression after chasing (*Appendix 1—figure 4*) in between rapid suppression outcomes and long-term chasing outcomes (*Appendix 1—figure 5*). Drive loss usually only occurred when migration values and low-density growth rates were both very low (*Appendix 1—figure 6*). At higher migration values and lower growth rates, 'suppression after chasing' became more limited in duration (*Appendix 1—figure 7*), and the average number of fertile females was reduced even during long-term chases (*Appendix 1—figure 8*). Such a reduction in females would likely substantially reduce disease transmission.

Overall, the three intermediate performance drives (zpg2, zpg2X, and nosF) had outcomes that depended heavily on the migration value and to a lesser extent on the low-density growth rate (*Figure 4*). The strongest drives, nosF2 and nosF3, were able to induce suppression over most of the parameter space, while a release of one of the three weakest drives, zpg, zpgX, and nos, usually resulted in long-term chasing outcomes.

## *Anopheles*-specific model

In addition to our discrete-generation model, we implemented a model that more explicitly simulates the expected dynamics of an *Anopheles* population by modeling overlapping generations using week-long time steps. The panmictic version of this model was used to calculate the genetic load of each drive as well as the speed at which it was able to spread, in the same manner as the discrete-generation model. The genetic load values in the discrete-generation model and the *Anopheles*-specific model were within 1% of one another for each drive parameterization; thus, only a single value is reported for each drive (*Figure 2*). The allele frequency trajectories varied slightly between the two models, but only for drives with X-shredders that biased the sex ratio (*Appendix 1—figure 1*). This occurred because the drive was mostly present in males, which have a shorter adult-stage lifespan than females, thus reducing the overall drive-allele frequency even though frequencies in emerging adults were the same.

The drives were next assessed in the spatial version of the *Anopheles*-specific model (*Figure 3B*). Generally, long-term chasing outcomes were more frequent in these simulations than in the discrete-generation model (*Figure 3A*). This came with an accompanying decrease in suppression outcomes as well as drive-loss outcomes. For example, in the discrete-generation model with a low-density growth rate of 10, zpg2 was able to suppress the population in 28% of simulations, while the drive suppressed in only 0.5% of simulations in the mosquito model. The average number of fertile females during chasing increased from 2567–4760. In the case of nosF, performance was similarly decreased from suppression in 54% of simulations to only 0.5% of simulations when the low-density growth rate was set to 10, with the average number of fertile females during chasing increasing from 2498–4101. In both models, nosF2 and nosF3 retained the ability to suppress the population. However, in the *Anopheles*-specific model, suppression often occurred only after an initial chase, and only nosF3 could reliably suppress the population in almost every simulation (*Figure 3B*). The zpg2X drive also performed more poorly in this model. In the discrete-generation model, the drive was able to suppress the population in 10% of simulations, but in the *Anopheles*-specific model, the drive was only able to suppress 0.5% of the time. The reduction in the number of fertile females achieved by both X-shredder drives in this

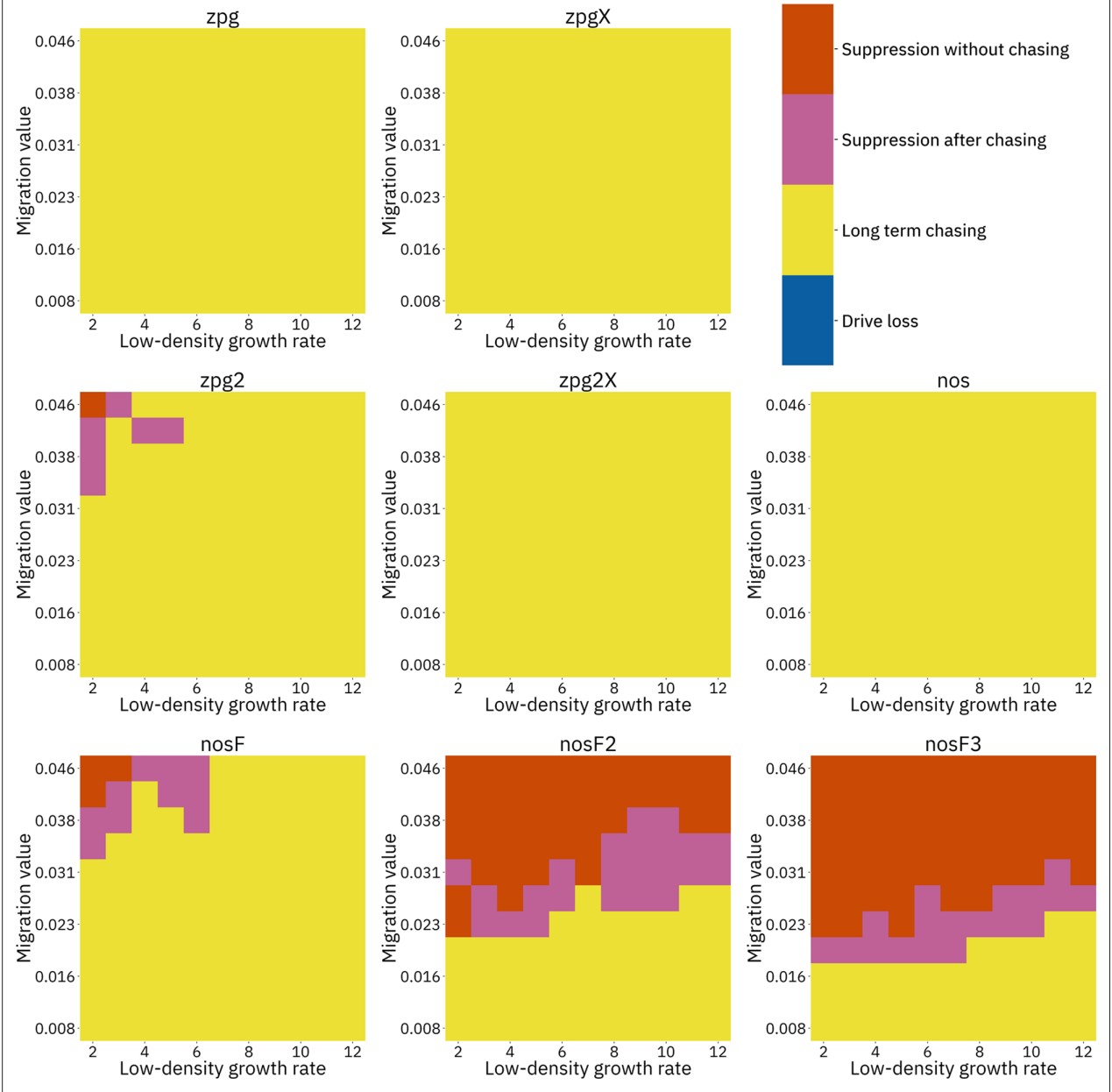

**Figure 5.** Impact of low-density growth rate and migration on outcomes in the *Anopheles*-specific model. The color of each square represents the outcome from among 20 simulations after adjustment to show the most representative outcome. The adjustment counts pairs of 'suppression without chasing' and 'long-term chasing' outcomes as two instances of 'suppression after chasing'. Note that the range of migration values in this model corresponds to the same net migration per generation as the range in *Figure 4* (see methods).

model was also less pronounced and, notably, was inferior for zpg2X as compared to zpg2. This was perhaps because the reduced genetic load of zpg2X outweighed the advantage of the male-bias. All the drives had similarly increased performance when the low-density growth rate was adjusted to 6 (*Appendix 1—figure 2B*). This allowed nosF2 to more reliably suppress the population, and the average number of fertile females was lower for all drives.

We next varied the low-density growth rate from 2 to 12 in steps of 1.0, while varying the migration value from 0.008 to 0.046 in steps of 0.0038 (thus corresponding to the same range that was examined in the discrete-generation model, see methods). The above tendencies mostly held true in this analysis as well (see *Figure 5* and *Appendix 1—figures 9–14*). Chasing outcomes tended to replace suppression outcomes and dominate large portions of the parameter space (*Appendix 1—figures*

*9–11*, compare with *Appendix 1—figures 3–5*). Drive-loss outcomes were extremely uncommon in this model compared to the discrete-generation model, with such outcomes occurring in significant quantity only when the low-density growth rate was very low (*Appendix 1—figure 12*, compare with *Appendix 1—figure 6*). Only nosF2 and nosF3 performed consistently well in this model and were able to suppress the population at moderate to high migration values (*Appendix 1—figure 9*, compare to *Appendix 1—figure 3*), or at least keep the duration of chasing and average number of fertile females at low levels (*Appendix 1—figures 13–14*, compare to *Appendix 1—figures 7–8*).

## Discussion

In this study, we analyzed several possible types of homing gene drives for population suppression of the malaria vector *Anopheles gambiae*. Various parameter sets for each of these drives were considered, representing alternative explanations for fertility measurements previously conducted (*Simoni et al., 2020*; *Kyrou et al., 2018*; *Hammond et al., 2021*). We found that the drive that combined a homing drive and X-shredder may actually be less capable of suppressing a population than the original *dsx* drive with the *zpg* promoter alone, particularly in a spatially continuous population. Further, we found that the *nos* promoter may be a preferable alternative for Cas9 expression compared to the *zpg* promoter, depending on which interpretation of the experimental data proves to be the most accurate.

A series of additional experiments could help ascertain which of our parameter sets most closely reflects the real-world dynamics of these drives. For example, the rate of paternal Cas9 deposition for a given Cas9 promoter could be assessed using a split-drive system (*Champer et al., 2019*), sequencing, or conducting a fertility assessment of offspring that have a drive-carrier father but that do not themselves inherit a drive allele. Though the exact rate may be somewhat different due to differences in expression at distant genomic loci, such an experiment should still serve to confirm the presence and magnitude of this phenomenon. Batch effects could also have a large impact on fertility measurements (*Hammond et al., 2021*), which we have also noticed in our *Drosophila* experiments (*Champer et al., 2020c*; *Champer et al., 2020a*; *Metzloff et al., 2022*; *Yang et al., 2022*). To address this, fertility experiments should, when possible, be performed with individuals that were raised in the same container (and then subsequently separated by fluorescence, ideally for multiple generations) and preferably even from the same parent (though this is only a readily available option for split-drive systems, especially when drive conversion efficiency is very high, as achieved by most *Anopheles* drives). Such experiments could potentially reveal that the negative fitness effects of the *nos*-Cas9 drive are smaller than initial results indicated. The *nos* promoter still needs to be assessed at the *dsx* target site, but this may prove to be an excellent combination if drive conversion remains high and resistance rates low. In general, further experimental work could also reduce uncertainty regarding these drives, even after the basic mechanisms are resolved.

Based on our modeling results, it appears that a combined X-shredder system may not represent an improvement to the standard homing suppression drive for a couple of reasons. First, with the alternate parameter set, the genetic load of this drive is lower (even with perfect homing, the 93% shredding rate would yield a genetic load of only 0.87) and is perhaps not high enough to reliably suppress a robust wild population. X-shredders fundamentally induce lower genetic load than female fertility homing drives when at the same germline efficiency, and a combined system can be expected to generally have the characteristics and genetic load of an X-shredder rather than a homing suppression drive, at least when the X-shredding efficiency is fairly high. Second, the drives including the X-shredder do not seem to perform well in spatial simulations, even with higher shredding rates due to drive wave dynamics and stochastic factors at high drive frequency (*Champer et al., 2021a*). This held true in our *Anopheles*-specific simulations, where the release of these drives usually resulted in long-term chasing with a high average number of fertile females. However, combined X-shredder systems are not without advantages. They can provide a way to support some continued suppression even in the presence of functional resistance alleles (although the rate that such alleles form at *dsx* is unknown and may already be sufficiently low due to the highly conserved nature of the target site, which is essential at the sequence level *Champer et al., 2021a*, and it could likely be further reduced by the use of a drive with multiplexed gRNAs *Champer et al., 2020b*). Also, because X-shredders bias the population toward males, such systems may effectively reduce biting females (and therefore

reduce disease transmission) more quickly than standard homing suppression drives (*Simoni et al., 2020*), although female *dsx* drive homozygotes also cannot bite (*Kyrou et al., 2018*).

Among the other drives we considered, the original zpg interpretation does not perform well due to paternal mosaic Cas9 deposition. This tends to greatly reduce genetic load because at high frequencies, female drive carriers (who suffer from paternal deposition) more often inherit their drive allele from a male drive parent because female homozygotes are also sterile. This reduces the maximum frequency that the drive can achieve, and thus its genetic load. Fortunately, the high rate of paternal deposition used to parameterize this variant is not likely to be an accurate interpretation of the data, especially considering the excellent performance of this drive in cage populations (*Kyrou et al., 2018*). Between the *nos* drives, the increased performance presumably comes from the reduction of somatic fitness costs between nos, nosF, nosF2, and nosF3. Though this does not affect the ultimate genetic load, it substantially affects the rate that these drives increase in frequency. In a spatial scenario, this factor allows the drive to more rapidly convert wild-type alleles into drive alleles, reducing the opportunities for a chasing dynamic to emerge, and allowing the drive to catch up more quickly if chasing does occur. The reasons for the slightly superior performance of nosF compared to zpg2 is less clear. Most likely the slightly increased genetic load and drive rate of increase account for this based on the slightly higher drive conversion and slightly lower germline resistance and fitness cost, despite the higher embryo resistance allele formation rate. Small variation in these parameters, perhaps even within the bounds of experimental error, could result in one or the other of these two drives coming out as superior to the other.

In general, the results of our *Anopheles*-specific continuous space model mirror the outcomes of our earlier generic model (*Champer et al., 2021a*) of suppression gene drives in continuous space, but the drives have substantially greater difficulty eliminating the population. Indeed, in the *Anopheles*-specific model, chasing outcomes were substantially more common, suggesting that chasing may be difficult to avoid when using a gene drive in natural populations unless the drive meets stringent efficiency criteria. These differences between models were likely at least partially due to the fact that competition in the *Anopheles* model affects offspring viability rather than female fecundity. Thus, fertile *Anopheles* individuals are not inhibited in reproduction due to competition by sterile individuals, as opposed to fertile females in the discrete-generation model. This ensures a more robust population even under pressure from a drive with a high genetic load when many sterile or otherwise non-contributing individuals (such as excess males due to an X-shredder) are present. The ability of the drive allele to migrate into wild-type populations is also reduced by way of females only mating once, and such reduced migration also tends to promote chasing (*Champer et al., 2021a*). Finally, the higher number of larvae generated by identical adult populations in the *Anopheles* model can reduce the chance of stochastic elimination as compared to the discrete-generation model, even in panmictic populations (*Liu et al., 2022*). Future studies could more precisely assess how different fundamental design decisions in continuous space models (e.g. the type of spatial competition and what stages of life competition occurs at) can impact the predicted outcome of different types of suppression drive releases.

It should be noted that while our *Anopheles* model represents a step toward increased realism compared to a discrete-generation model, our study still has various limitations. For example, distances were unitless (although see Materials and methods for possible comparison to previous field studies), and higher relative dispersal rates in real populations may enable easier suppression. We did not consider the possibility of long-distance migration, which could make chasing more likely by bringing distant wild-type individuals directly into empty areas cleared by the drive, or functional resistance, which can evolve more readily during lengthy chases (*Champer et al., 2021a*). We also did not take into account other factors that may be important in real-world populations, such as seasonality, interspecies competition (*Liu et al., 2022*), variation in survival rates due to climate or predation, heterogeneous landscapes, nonrandom movement, genetic variation, or the possibility of various types of evolutionary responses to the spread of a suppression drive in the population (*Champer et al., 2021a*; *Bull et al., 2019*; *Gomulkiewicz et al., 2021*; *Cook et al., 2022*). Our parameterization of the drives was limited by our input data, with small sample sizes in particular causing high error in our estimates of nonfunctional resistance allele formation rates. In general, any modeling study is limited in the level of detail it can provide for predicting the outcome in any particular real-word scenario, especially a scenario as complex as a

gene drive release. However, increasing the level of detail present in a model can potentially allow for more accurate predictions. At the very least, modeling studies can identify a possible range of outcomes, which can then provide a useful framework from which to consider possible gene drive deployment options.

One potentially important implication of these models is that many of the gene drive designs currently under consideration may be quite close to the boundary between drives capable of successful suppression and drives that could be expected to fail due to chasing. For such borderline drives, the outcome of a drive release could be very sensitive to the precise ecological characteristics of the targeted population. Small parameter differences in drive performance could therefore be critical in ensuring drive success, suggesting that currently ambiguous drive characteristics should be thoroughly considered, and all drive parameters should be measured with as much accuracy as possible. At the same time, computational models must be further refined in order to improve their predictive accuracy, such that we can more reliably assess whether a candidate suppression drive indeed meets any specific requirements for success, or to at least understand the most likely outcome.

In conclusion, our modeling indicates that several promising candidates for suppression gene drives could have substantially different overall effectiveness in spatial mosquito populations than predicted by panmictic population models. Specifically, we believe that a standard homing suppression drive using the *nos* promoter (*Carballar-Lejarazú et al., 2020*; *Hammond et al., 2021*) for Cas9 (possibly a high fidelity variant to avoid off-target cleavage *Langmüller et al., 2022*) and targeting *dsx* (*Simoni et al., 2020*; *Kyrou et al., 2018*) with two or more closely spaced gRNA target sites (*Yang et al., 2022*; *Champer et al., 2020b*) may be the optimal combination of currently available tools for population suppression of *Anopheles* mosquitoes.

## Materials and methods
### Gene drive mechanisms

All of the gene drives modeled in this study are designed to target a female fertility gene that is essential but haplosufficient (*Figure 1*). Cas9 is directed by a guide RNA (gRNA) to cleave a specific target site located within this gene. Through homology-directed repair, the drive allele then can copy and paste itself into the fertility gene in a manner that effectively inactivates the gene. Since this target gene is haplosufficient, female drive heterozygotes would be fully fertile if Cas9 activity is restricted to the germline (yet most variants we model also have somatic expression that can reduce fertility, see below). However, once the drive allele has spread to a high frequency in the population, an accumulation of sterile drive-homozygous females will cause the population to be reduced or to completely collapse.

If the cleaved target site does not undergo homology-directed repair but instead is repaired by the error-prone process of end-joining, the resulting mutations may render the site unrecognizable to future Cas9 cleavage. Most of the time, such resistance alleles do not preserve the function of the target gene (these are known as 'r2 alleles') and thus generally do little more than slow the spread of the gene drive. A more severe problem is posed by 'r1' resistance alleles, which preserve the function of the target gene and can thereby prevent the suppressive effects of the drive. These r1 alleles were not observed in a cage study of a drive that targeted a highly conserved site (*Kyrou et al., 2018*), and the use of multiplexed gRNAs could also limit the formation of such alleles (*Champer et al., 2020b*). We assume that one or both of these methods will be effective and therefore only model 'r2' resistance alleles in our simulations.

One of our drives is motivated by a recent study in which a female fertility homing drive and an X-shredder were combined at the same genomic locus (*Simoni et al., 2020*). The X-shredder is designed to cleave multiple sites on the X-chromosome in gametes, rendering them nonviable. This biases the sex ratio towards males because most viable gametes will carry a Y-chromosome. Female gametes that escape destruction may still be sterile since the homing drive disrupts a female fertility gene target as before. Overall, the X-shredder reduces the number of drive females, thereby reducing the influence of drive-based somatic fitness costs in females. Unlike a Driving-Y design, this autosomal X-shredder cannot increase its own inheritance. It relies on the linked homing drive element for this purpose.

## Modeling of gene drive

In our model, gene drive processes occur independently in each gametocyte of each reproducing individual (*Figure 1*). A wild-type allele in a drive heterozygote is converted into a resistance allele with a probability equal to the germline resistance allele formation rate, which differs by sex of the individual. We assume that all resistance alleles are 'r2' alleles that disrupt the function of the target gene. If the allele remains wild-type, it is converted to a drive allele at a rate equal to the drive conversion rate, which also differs by sex.

We next model the further potential for resistance to form due to maternal Cas9 deposition into the embryo. This process converts a wild-type allele inherited from a drive-heterozygous parent into a resistance allele with a probability equal to the maternal embryo resistance allele formation rate. In two drive parameterizations, we model paternal Cas9 deposition as well. In the case of paternal deposition, we assume the effects are mosaic based on experimental interpretation (thus only reducing female fertility), rather than affecting all cells in the new offspring. In this case, germline drive conversion still occurs, assuming the individual is drive-heterozygous, but females that receive paternally deposited Cas9 are rendered sterile.

In this study, we considered drives based on the *nos* and *zpg* promoters for Cas9. The drive could also have an X-shredder (zpgX and zpg2X). However, due to diverging interpretations of existing experimental data, we modeled four possible variations of *nos* drives and two possible interpretations of *zpg* drives. We term these variations nos (without italics), nosF, nosF2, nosF3 (with the latter versions having a lower somatic fitness costs) and zpg (without italics), zpg2, zpgX, and zpg2X (with zpg2 and zpg2X interpreted as not having paternal deposition of Cas9, and instead having a higher somatic fitness cost in females). See the results and *Table 1* for explanations and parameterizations of each of these drives.

If the gene drive includes an X-shredder and the father has a drive allele, then the probability that a given offspring will be male is equal to $\left[\frac{1}{2-(\text{X shredding rate})}\right]$. Thus, if Cas9 shreds the X-chromosome of gametes in males 100% of the time, all of a male's offspring will be male, and if the X shredding rate is 0, then the probability that an offspring is male is the usual 50%. The presence of an X-shredder only serves to bias the sex ratio of the offspring, and is not considered to reduce the number of offspring that an individual is expected to leave.

## Discrete-generation panmictic simulation model

See *Appendix 1—table 1* for a list of key parameters of all models. The discrete-generation model is based on an earlier study (*Champer et al., 2021a*) and simulates a population of 50,000 sexually reproducing diploids with non-overlapping, discrete generations using the forward genetic simulation software SLiM (version 3.7; *Haller and Messer, 2019*). The wild-type population is allowed to equilibrate for 10 generations before gene drive heterozygotes are released at a frequency such that they represent 1% of the population.

In the panmictic discrete-generation model, each fertile female randomly selects a mate. We then evaluate the fecundity of the female, which can be reduced by the fitness cost of somatic Cas9 expression. Fecundity is also reduced if the male mate is a nos drive heterozygote (though not in the nosF, nosF2, and nosF3 drives).

Female fecundity ($w_i$) is further scaled according to a Beverton–Holt model by how close the population size ($N$) is to the carrying capacity of the system ($K$): $w_i' = w_i \times \frac{\beta}{(\beta-1)(\frac{N}{K})+1}$, where $\beta$ is equal to the low-density growth rate of the population. We determine the number of eggs produced by drawing from a binomial distribution, with 50 trials and a success probability equal to $\frac{w_i}{25}$ such that a wild-type female is expected to produce 2 offspring on average when the population is at capacity.

Simulations were run until the drive was lost, the population was eliminated, or 1000 generations had elapsed if neither event occurred. In some simulations, modifications were made to facilitate accurate measurement of genetic load (see supplemental methods).

## Discrete-generation spatial model

We extend our panmictic model into continuous space by explicitly tracking every individual's position across a 1 × 1 (unitless) area, similar to a model we introduced in a previous study (*Adolfi et al., 2020*). The simulation begins with 50,000 wild-type individuals that are randomly distributed across the landscape. After 10 generations, a number of drive-heterozygous individuals representing

1% of the total population are released from a 0.01-radius circle at the center of the arena. In the reproduction stage, fertile females can only sample potential mates from a circle surrounding the female with a radius equal to the migration value parameter (with a default of 0.04). If there are no males in this circle, then the female does not reproduce. Reproducing females have a fecundity of $w_i' = w_i \times \frac{\beta}{(\beta-1)(\frac{\rho_i}{\rho})+1}$, where $\rho_i$ is the local density in a 0.01-radius surrounding the female and $\rho$ is the density value that would be expected if the population were evenly distributed across the landscape. This means that a female in a low-density area will have greater fecundity, reflecting greater access to resources as well as reduced competition faced by her offspring. Each offspring is displaced a random distance from its mother. Displacement distance in the $x$ and $y$ axis are both drawn from a normal distribution with a mean of 0 and standard deviation equal to the migration value parameter. This produces an average displacement of migration value $\times \sqrt{\pi/2}$. If an offspring's coordinates fall outside the bounds of the simulation, the coordinates are redrawn until the offspring is placed within the boundaries.

During each simulation, we calculated Green's coefficient, which provides a quantification of the degree of spatial clustering. To do so, we divided the 1 × 1 area into an 8 × 8 grid and counted the number of individuals present in each of the 64 grid sections. Green's coefficient ($G$) is then defined by $G = \frac{\frac{s^2}{n}-1}{N-1}$, where $N$ is the total population size, $n$ is mean number of individuals in a grid section, and $s^2$ is the variance of the counts. If individuals are distributed randomly and according to a Poisson distribution, then it is expected that $n = s^2$ and $G = 0$. By contrast, if all individuals are maximally clustered into a single section of the grid, $G = 1$. Note that we only count wild-type homozygotes in this measurement, because this was found to provide a more useful representation of the spatial dynamics (*Adolfi et al., 2020*).

As observed previously (*Champer et al., 2021a*), the release of a suppression drive can result in chasing dynamics between the drive and wild-type alleles. When a suppression drive is first released from the center of the landscape, the population is suppressed radially outward, such that surviving wild-type individuals cluster near the boundaries. This causes Green's coefficient to increase. However, wild-type individuals that escape into areas previously cleared by the drive are able to produce more offspring as a result of the lack of competition in these areas. If a wild-type population grows into a previously empty area, then Green's coefficient once again decreases as these individuals occupy more territory. We aimed to capture this inflection point by finding the first local maximum in Green's coefficient and the first local minimum in the number of wild-type alleles. These events tend to occur within 5 generations of one another. To define the generation when chasing begins in a simulation, we chose the earlier of these two time points.

### *Anopheles*-specific model

In addition to the discrete-generation models, we implemented refined versions of the models that more explicitly simulate the life-cycle, demography, and ecology of *Anopheles* mosquitos. These models progress by weekly time-steps, allowing for overlapping generations.

Female mosquitoes of most species usually just mate once, though older females have often been observed to mate a second time (*Degner and Harrington, 2016*; *Richardson et al., 2015*; *Tripet et al., 2003*; *Gomulski, 1990*). Females that have already mated store sperm from the male, which is used to continue to fertilize eggs in future weeks. This behavior is implemented in the model. We implemented a 5% chance each week that the female will re-mate, resulting in the new mate fathering any future offspring unless re-mating occurs again. This 5% weekly re-mating value is based on estimates from *A. gambiae* experimental (*Gomulski, 1990*) and field (*Tripet et al., 2003*) studies, assuming lower field survival rates.

After reaching adulthood, females have a 50% probability to successfully produce offspring during any given week if they have previously mated (which takes place before offspring production). The number of eggs laid is not density dependent but is instead drawn from a Poisson distribution with the average set at 50 times the product of the fitness of the two parents. The number of eggs laid by *A. gambiae* appears closer to three times this level in laboratory conditions (*Yaro et al., 2006*), but in practice, usually only a far smaller number reach adulthood in wild conditions. Our use of 50 allows larvae at low density to have high survival rates compared to most wild conditions while still minimizing computational burden.

The survival rates of adults are also not density dependent. *Anopheles* females have longer lifespans than males. In our model, males never survive beyond their third week as an adult and females never beyond their sixth, with the survival rates at each age of adulthood given as follows:

$$\text{Adult male survival rates}: \quad \left[\frac{2}{3}, \frac{1}{2}, 0\right]$$

$$\text{Adult female survival rates}: \quad \left[\frac{5}{6}, \frac{4}{5}, \frac{3}{4}, \frac{2}{3}, \frac{1}{2}, 0\right]$$

This results in an approximately linearly declining number of surviving adult members of a single week cohort. This function was chosen based on survival curves in laboratory studies (*Christiansen-Jucht et al., 2014*) and to allow simulation of age-based health. The measured male survival rate in the field was measured as approximately 30% per week in one recent study (*Yao et al., 2022*), and often higher in females and closer to 50% in males for older studies (*González Jiménez et al., 2019*; *Matthews et al., 2020*). This is somewhat lower, but broadly similar to our model, considering high variation in field conditions and our need to simulate a high effective population size with limited computational resources.

Note that individuals have the opportunity to mate and reproduce before these survival rates toll. One generation in the discrete model is then equivalent to ~3167 weeks in this model (thus, we ran these simulations for 3167 weeks to represent 1,000 generations).

Mosquito larvae often face fierce competition for resources in the small bodies of water in which they develop while adults do not directly compete with one another for food (*Arifin et al., 2014*). Thus, population density does not regulate adult fecundity in our *Anopheles* model; instead, it affects juvenile survival. In our model, individuals are considered to be in juvenile stages (egg, larvae, and pupae) during their first two weeks of life and reach adulthood when they enter their third week. The larger week-old larvae do not compete with new eggs, and they cease to compete completely once they reach the pupal stage. However, these larger larvae consume more resources compared to smaller larvae and are thus estimated to exert competition at fivefold strength compared to new juveniles. In the panmictic version of this model, newly generated individuals survive until adulthood with a probability that depends on the global sum of new juveniles $n$, the global sum of week-old larvae $o$, the low-density growth rate $\beta$, as well as the expected competition within the system, which is in turn a function of the number of adult females in a population at capacity (25,000 in all simulations) as well as the expected number of offspring per adult female per week (25), calculated as follows:

$$\text{expected competition} = 25000\left(25 + 2 \times 0.285714 \times 5\right)$$

$$\text{competition ratio } r = \frac{(n + 5o)}{\text{expected competition}}$$

$$\text{new offspring survival rate} = \frac{\beta}{25\left((\beta - 1)\, r + 1\right)} \times \left[\frac{1}{2 \times 0.285714}\right]^{-r}$$

Here, 0.285714 adjusts for the expected number of older juveniles when the population is at equilibrium such that the adult female population size remains at the specified equilibrium value of 25,000. Note that while juvenile mosquitoes experience continuous mortality in real populations, our model approximates this by determining total juvenile mortality immediately after new juveniles are all generated, representing mortality in both 1-week-old and older juveniles (all juveniles that survive this stage will survive to adulthood). This allows more individuals to be culled immediately, thus reducing the computational burden of evaluating the large number of spatial interactions determining competition, thereby allowing for the simulation of larger populations. This approximation is supported by the fact that larvae in their second week are much larger than newly hatched larvae, thus preventing younger larvae from substantially reducing the resources available to older larvae. Later in their second week, the juveniles are pupae, which do not require additional resources. Thus, new juveniles are not likely to affect the mortality of juveniles that are at least a week older, supporting our approximation of determining mortality only in the first week.

### *Anopheles*-specific spatial model

In the spatial version of this model, the survival rate of new offspring is affected by the local density of other larvae, rather than global counts thereof. The amount of competition experienced is determined by other new offspring and week-old larvae nearby. The maximum amount of competition contributed by other new offspring is 1.0, which linearly declines to 0.0 at a distance of 0.01 (the average competition contributed by another individual within range is therefore 1/3). Week-old larvae contribute five times as much competition. The expected competition again corresponds to the number of females when the population is at capacity, as well as the expected number of offspring per female:

$$\text{expected competition} = \frac{25000 \left(25 + 2 \times 0.285714 \times 5\right)}{3\pi \times 0.01^2}$$

After determining the amount of competition being faced by a new offspring as compared to the expected competition, the survival rate of that new offspring is set in the same way as in the panmictic version of the model.

In the *Anopheles* spatial model, surviving adults migrate in the same manner that new offspring are dispersed from their mothers, which is implemented identically to the discrete spatial model except that the migration value now represents the average displacement per week, with the default of 0.0307 creating an equal displacement per generation (males and females together averaging 2.67 displacements in a generation) as well as an equal "drive wave advance speed" to the discrete generation model default of 0.04 (with an average displacement of 0.05).

Comparing our default dispersal rate of 0.307 to a recent study that found a mean dispersal of 171 m over 20 days (*Yao et al., 2022*), we can potentially state that our mean dispersal corresponds to 101 m per week and that our simulation area length is therefore 3.3 km. A population density of 430 mosquitoes per hectare would thus yield a total population of 470,000 adults, compared to our default of 40,000 (of which 25,000 are females). However, the effective population (more akin to what is generally analyzed in population genetic models) would be substantially below 470,000, and there is substantial density variation between wild mosquito populations. Thus, our modeled population density could be considered reasonable when modeling *Anopheles*, based on length scales from our migration rate. However, it is unclear what parameter value for the density interaction radius would best represent actual mosquito populations, given the varying size and distribution of larval habitat, as well as larval movement. Thus, our value of 0.01 stands in as an estimate, chosen as a low value that is still large enough to avoid extreme variation in individual larval competition when the population is at equilibrium before a drive release.

### Data generation

Simulations were run on the computing cluster at the Department of Computational Biology at Cornell University. Data processing and analytics were performed in Python, and figures were prepared in Python and R. All SLiM files for the implementation of these suppression drives are available on GitHub (https://github.com/jchamper/ChamperLab/tree/main/Mosquito-Drive-Modeling, copy archived at swh:1:rev:bff233aa54cd8f9b94151660a8de98adeda92c33, *Champer, 2022*).

## Acknowledgements

This study was supported by the National Institutes of Health award F32AI138476 to JC and award R01GM127418 to PWM.

## Additional information

#### Competing interests

Philipp W Messer: Reviewing editor, *eLife*. The other authors declare that no competing interests exist.

## Funding

| Funder | Grant reference number | Author |
|---|---|---|
| National Institutes of Health | F32AI138476 | Jackson Champer |
| National Institutes of Health | R01GM127418 | Philipp W Messer |

The funders had no role in study design, data collection and interpretation, or the decision to submit the work for publication.

## Author contributions

Samuel E Champer, Isabel K Kim, Data curation, Software, Formal analysis, Validation, Investigation, Visualization, Methodology, Writing – original draft, Writing – review and editing; Andrew G Clark, Philipp W Messer, Supervision, Project administration, Writing – review and editing; Jackson Champer, Conceptualization, Data curation, Formal analysis, Supervision, Funding acquisition, Investigation, Methodology, Writing – original draft, Project administration, Writing – review and editing

## Author ORCIDs

Samuel E Champer http://orcid.org/0000-0002-4559-7627
Isabel K Kim http://orcid.org/0000-0001-8877-5174
Philipp W Messer http://orcid.org/0000-0001-8453-9377
Jackson Champer http://orcid.org/0000-0002-3814-3774

## Decision letter and Author response

Decision letter https://doi.org/10.7554/eLife.79121.sa1
Author response https://doi.org/10.7554/eLife.79121.sa2

# Additional files

## Supplementary files

• MDAR checklist

## Data availability

All SLiM files for the implementation of these suppression drives are available on GitHub (https://github.com/jchamper/ChamperLab/tree/main/Mosquito-Drive-Modeling, copy archived at swh:1:rev:bff233aa54cd8f9b94151660a8de98adeda92c33).

The following dataset was generated:

| Author(s) | Year | Dataset title | Dataset URL | Database and Identifier |
|---|---|---|---|---|
| Champer SE, Kim IK, Clark AG, Messer PW, Champer J | 2022 | Mosquito-Drive-Modeling | https://doi.org/10.5281/zenodo.7233289 | Zenodo, 10.5281/zenodo.7233289 |

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

# Appendix 1

## Supplemental Information

Supplemental Methods

## Measurement of genetic load in panmictic simulations

A subset of panmictic simulations was used to assess the genetic load of the drives (the reduction in reproductive capacity of the population compared to a population that is identical except for being composed entirely of wild-type individuals). For sterility-based homing suppression drives, this measures the reduction in fertility caused by the presence of the drive. As measured at any time $t$ in the simulation, the genetic load is a function of the number of females ($N_t$) alive at time $t+1$ and the number that could be predicted if there were no drive present:

$$N_{f, t+1, \text{predicted}} = N_f \frac{\beta}{(\beta - 1)\left(\frac{N_t}{K}\right) + 1}$$

$$\text{genetic load}_t = 1 - \frac{N_{f,t+1, \text{actual}}}{N_{f,t+1, \text{predicted}}}$$

However, genetic load as measured in this manner is merely an instantaneous measurement – it describes the impact a drive has on the population over the course of only a single time step. Further, it is not always possible to simply average the genetic load measurements from several time steps in a row because the population can cease to exist under the pressure of the drive before sufficient measurement can be made. To remedy this, in simulations specifically seeking to determine genetic load, fertile females had a multiplier applied to the number of offspring they generated in order to approximately maintain the population at capacity given the number of infertile individuals in the system at any given time. Specifically, the number of offspring produced by fertile females was divided by the following population factor:

$$\text{population factor} = \left(\frac{\text{fertile males}}{N_m}\right)\left(\frac{\text{fertile females}}{N_f}\right)\left(\frac{N_f}{(N_m + N_f)}\right) \times 2$$

Similarly, the numerator term of the measured genetic load was multiplied by this factor in order to account for the excess individuals beyond the number that would normally be present in the next time step.

**Appendix 1—table 1.** Model Demographical/Ecological Parameters.

**All models**
Low density growth rate = 6 or 10 (or varies from 2 to 12)
Release amount = 500 new adult male heterozygotes
Competition distance = 0.01
Drive release radius = 0.1

**Discrete generation model**
Time step = one generation
Capacity = 50,000
Maximum offspring = 50
Migration and mating distance = 0.04 (or varies from 0.01 to 0.06)

*Appendix 1—table 1 Continued on next page*

Appendix 1—table 1 Continued

**Anopheles model**
Time step = one week
Adult female capacity = 25,000
Female remate chance = 0.05
Fraction of females that reproduce per week = 0.5
Average offspring in one batch = 50
Old larva relative competition contribution = 5
Migration and mating distance = 0.0307 (or varies from 0.008 to 0.046)
Adult female mortality rates (listed by age) = [5/6, 4/5, 3/4, 2/3, 1/2, 0]
Adult male mortality rates (listed by age) = [2/3, 1/2, 0]

## Supplemental Results

### Parameterization of drive candidates

We first examined drive inheritance from *zpg* promoter drive heterozygous individuals from two studies (*Kyrou et al., 2018*; *Hammond et al., 2021*). Discounting males that did not show biased inheritance (it is likely that resistance formed in such males while they were embryos due to maternally deposited Cas9), the drive conversion rate appears to be 96% for male heterozygotes and 99% for female heterozygotes. We set the germline resistance allele formation rate to be 2% for males and 1% for females so that approximately half of the alleles that fail to undergo drive conversion remain wild-type (these estimates don't substantially impact our results since we assume that all resistance alleles disrupt the function of the drive). Males with drive mothers (who therefore receive maternally deposited Cas9 and gRNA) that failed to show substantial drive conversion in the germline likely had embryo resistance alleles, as would sterile females with drives mothers. We therefore posit an embryo resistance allele formation rate of approximately 8% based on these frequencies. The authors hypothesized that reductions in female fertility could also be due to paternal deposition leading to resistance allele formation (*Hammond et al., 2021*). Based on the rate of sterile female drive carriers with male drive parents, we calculated the paternal resistance allele formation rate as 69%. However, unlike maternal resistance alleles, we consider these paternal resistance alleles to be mosaic at a sufficient level to cause complete female sterility, but not present at sufficient level in the germline to prevent efficient drive conversion (thus, these resistance alleles have no effect on male progeny, in line with experimental data). We then applied a 30% fitness cost in all drive/wild-type heterozygous females to account for the remainder of the reduced fertility of drive females compared to wild-type females, though this reduction could also be interpreted as additional mosaic parental deposition from both drive mothers and fathers.

However, alternative explanations exist for the reduced fertility in female drive heterozygotes with male drive parents as compared to those with female drive parents. These reduced fertility measurements could be the result of batch effects or simply be random variations. If interpreted in this manner, these fertility reductions can be explained by somatic Cas9 expression and cleavage, with no significant paternal deposition. This is supported by another data set with a different target gene but identical drive components, wherein female heterozygotes had similar fertility regardless of which parent provided the drive allele (*Hammond et al., 2021*). Fundamentally, it seems unlikely that paternal deposition causes more embryo resistance allele formation than maternal deposition, considering the relative size of the sperm and the egg. The fact that the zpg drive targeting *dsx* completely suppressed a cage population is also indirect evidence against the drive being so negatively impacted by paternal deposition, as we found that this interpretation of the drive has a relatively low genetic load, which would perhaps be insufficient to suppress a cage population (see *Table 1*). Specifically, a single fully fertile female mosquito appeared to be able to generate 130 viable larval progeny based on individual crosses (*Kyrou et al., 2018*). This could mean that 650 eggs (the number used for each generation of the cage) could potentially be generated by as few as five fully fertile females. Even conservatively assuming that 650 eggs yields only 150 adult females, a genetic load of at least 0.97 in a deterministic model is needed to reduce the viable egg count of the next generation to below 650 on average. Though stochastic effects could certainly result in a cage population being suppressed by a drive with a somewhat lower genetic load, stochastic variation represents an increasingly unlikely explanation at lower genetic loads. Finally, previous studies have found no indication of paternal Cas9 deposition in *Drosophila* (*Champer et al., 2019*; *Champer et al., 2020b*; *Champer et al., 2018*; *Champer et al., 2017*) and nothing conclusive

in *Anopheles*. Evidence of the phenomenon found in *Anopheles* studies (*Hammond et al., 2016*; *Galizi et al., 2016*) could also potentially be similarly explained as effects caused by leaky somatic Cas9 expression (though significant paternal deposition has been clearly observed with a different nuclease *Galizi et al., 2014*; *Windbichler et al., 2008*). In our models, zpg2 is parameterized to match this alternative explanation. Instead of a 69% paternal embryo resistance allele formation rate, this implementation has a 50% fitness cost from somatic cleavage in female heterozygotes (increased from 30%, thus matching the average fertility reported in two studies *Kyrou et al., 2018*; *Hammond et al., 2021*).

The *nos* promoter represents a potential alternative to *zpg*'s germline-restricted expression of Cas9. We parameterize this drive based on a previous study (*Hammond et al., 2021*) that involved targeting the *nudel* gene. Thus, a *nos*-based drive at *dsx* may actually have somewhat different performance parameters (*zpg* based drives were slightly more effective at *dsx Kyrou et al., 2018* than at *nudel Hammond et al., 2021*). Here, we parameterize this drive with a 99% drive conversion in female drive/wild-type heterozygotes, 98% drive conversion in males, 1% germline resistance allele formation for both sexes, and a 14% embryo resistance allele formation rate in the progeny of female drive heterozygotes. In the previous study, the *nos* promoter was less favored than *zpg* because fertility was reduced by 45% in crosses between drive heterozygous females and wild type males as well as in crosses between drive heterozygous males and wild type females (*Hammond et al., 2021*; ). We model this as a fitness cost from leaky somatic expression and cleavage of Cas9 in both male and female heterozygotes. However, batch effects may have influenced fertility determination in this study, because the wild-type controls for each promoter themselves had substantial differences, Further, there is no clear explanation for somatic cleavage negatively affecting male heterozygotes. We therefore also model an alternative interpretation of the nos drive in which there is no somatic fitness cost in males (which we term nosF), which we consider is likely a more accurate representation of this drive. If we further assume that *nos* male drive heterozygotes do not have a fitness cost from somatic expression and can serve as a control for female drive heterozygotes, then only a 15% fitness reduction from somatic expression in the females (together with the same 14% embryo resistance allele formation rate in the progeny of female drive heterozygotes) can explain their relative fertility. We consider this as another alternative possible parameter set (nosF2). We also model a fourth drive in which there are no somatic fitness costs in either sex, which is within the margin of error of the fertility experiment, though we consider this a highly optimistic parameter set (nosF3).

We note that all performance parameters for the nosF drive are very similar to the zpg2 drive parameterization. While sample sizes for drive conversion and for fitness calculations are fairly high, our estimate of germline resistance allele formation is based on common modeling convention rather than robust experimental evidence, and our estimate for the embryo cut rate are based on modest sample sizes. Thus, the exact difference between these drives is somewhat uncertain and would require additional experimentation to confirm. Other differences between our drives' parameter sets are based on qualitative assumptions that provide a clearer contrast for interpretation of results.

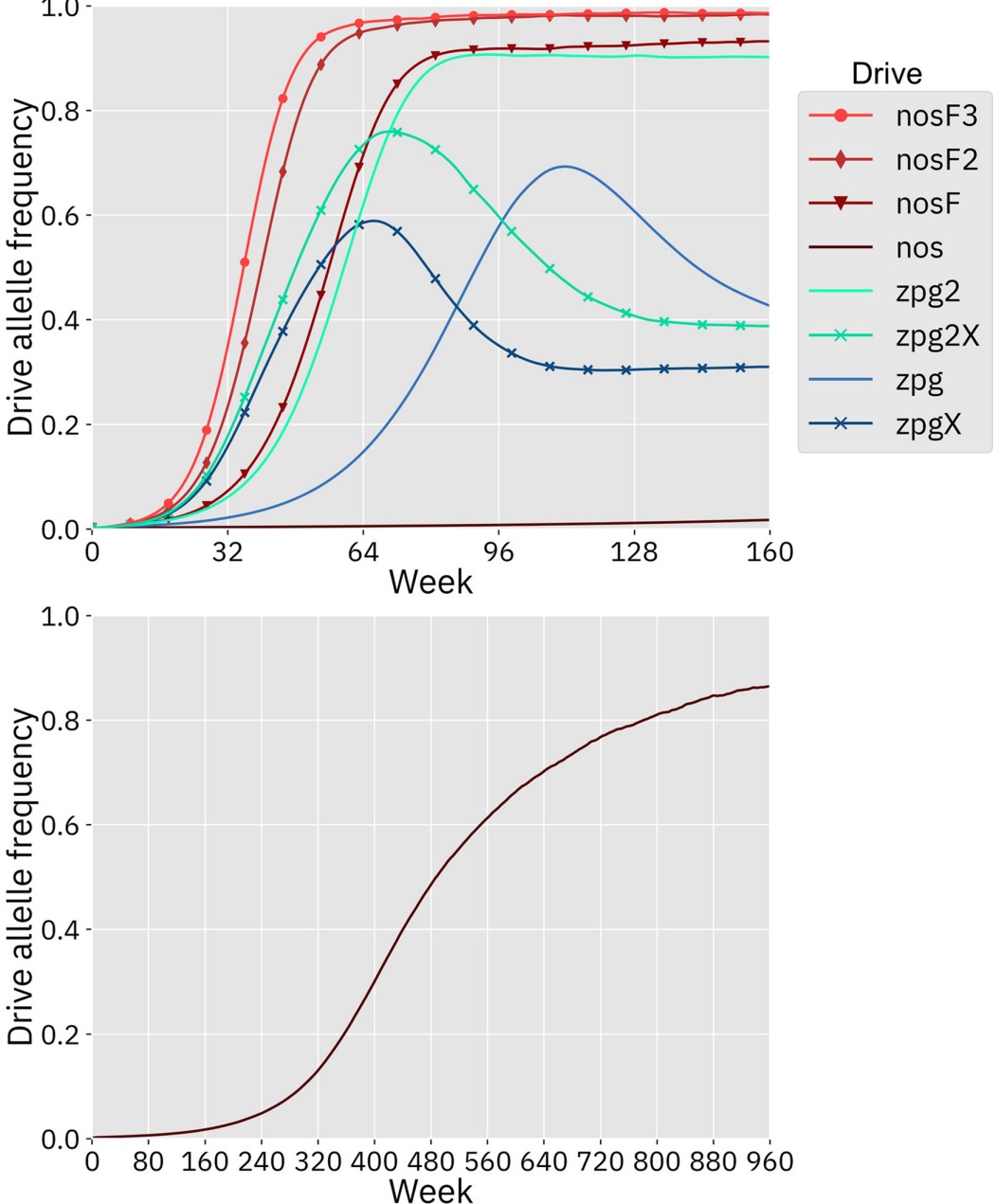

**Appendix 1—figure 1.** Drive allele frequency trajectories in the *Anopheles*-specific model. Using default parameters and with 20 replicates per drive, each drive was released into a panmictic population. The average allele frequency for each week is displayed. Offspring were artificially generated from fertile individuals at high rates to prevent complete population suppression even at high drive frequencies and genetic loads (see Supplemental Methods). The nos drive increases more slowly at low frequency, but eventually reaches a high equilibrium frequency, as seen in the lower panel.

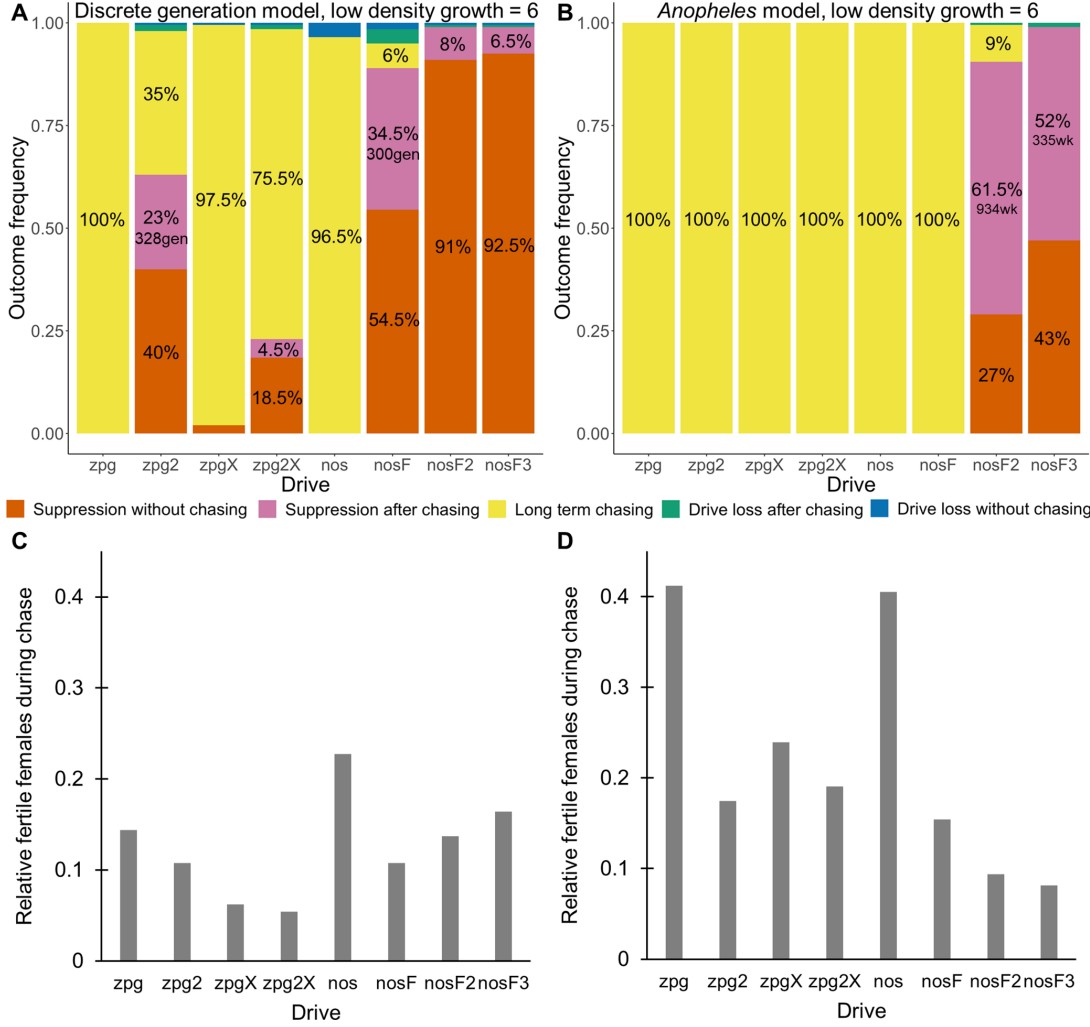

**Appendix 1—figure 2.** Outcomes in the spatial models with reduced low-density growth rate. Using default parameters, a low-density growth rate of 6, and with 200 replicates per drive, each drive was released into the middle of a wild-type population. The outcome was recorded after 1000 generations or when the population was eliminated for the discrete-generation (**A**) and *Anopheles*-specific (**B**) models. In outcomes involving chasing followed by suppression, the number of generations (gen) or weeks (wk) between the start of chasing and population elimination is shown. Also displayed is the relative number of fertile females during periods of chasing (including both long-term and short-term chasing outcomes) compared to the starting amount prior to release of the drive for the discrete-generation (**C**) and *Anopheles*-specific (**D**) models. Due to the high number of replicates, the error for each data point is negligible, except for the nosF2 and nosF3 drives in the discrete-generation model due to the short duration of chasing.

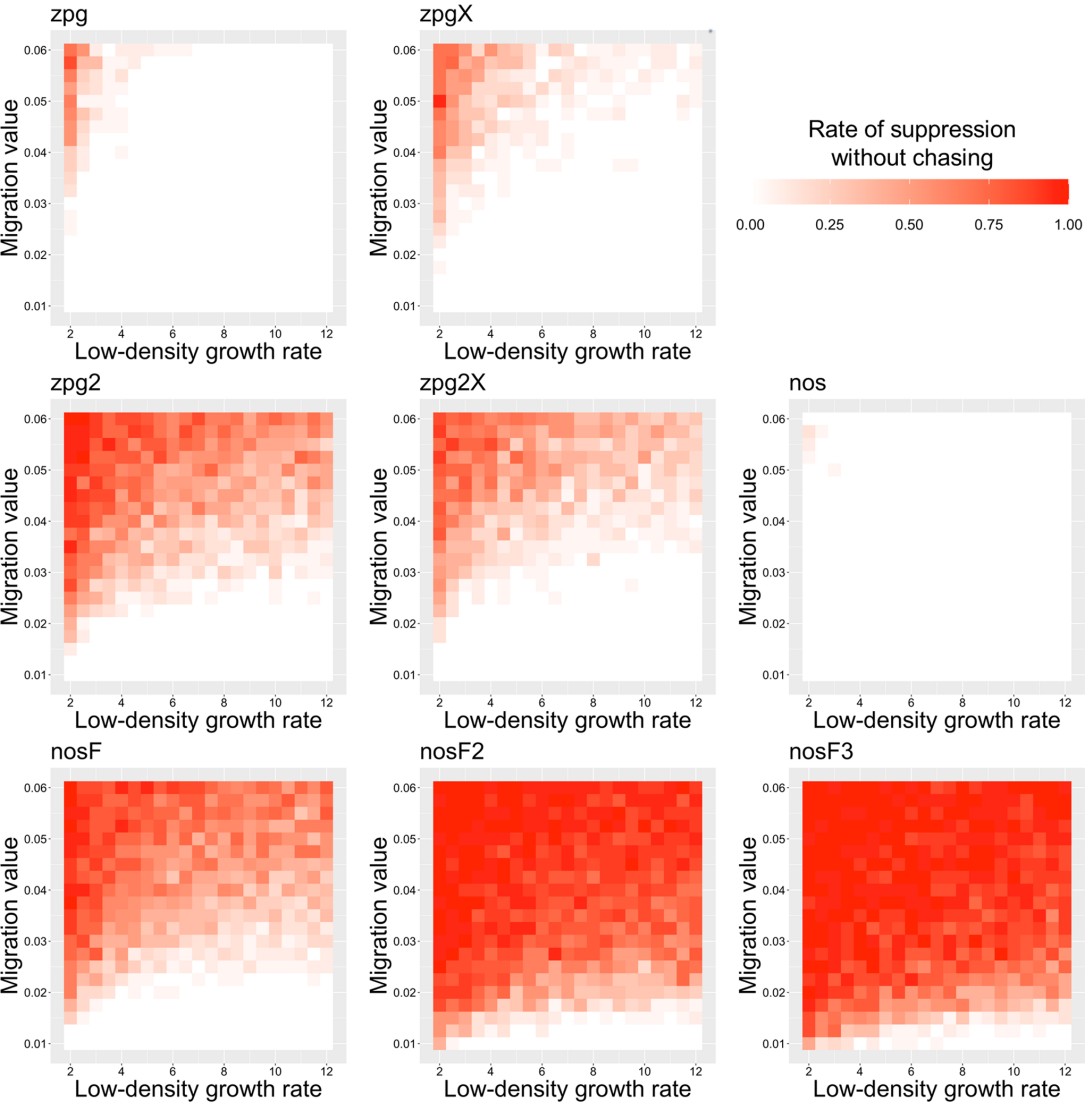

**Appendix 1—figure 3.** The rate of suppression without chasing in the discrete-generation model. Drive-carrying individuals were released into the middle of a wild-type population. The proportion of simulations in which suppression occurred either before a chase or within 10 generations of the start of chasing is shown. Each point represents the average of 20 simulations.

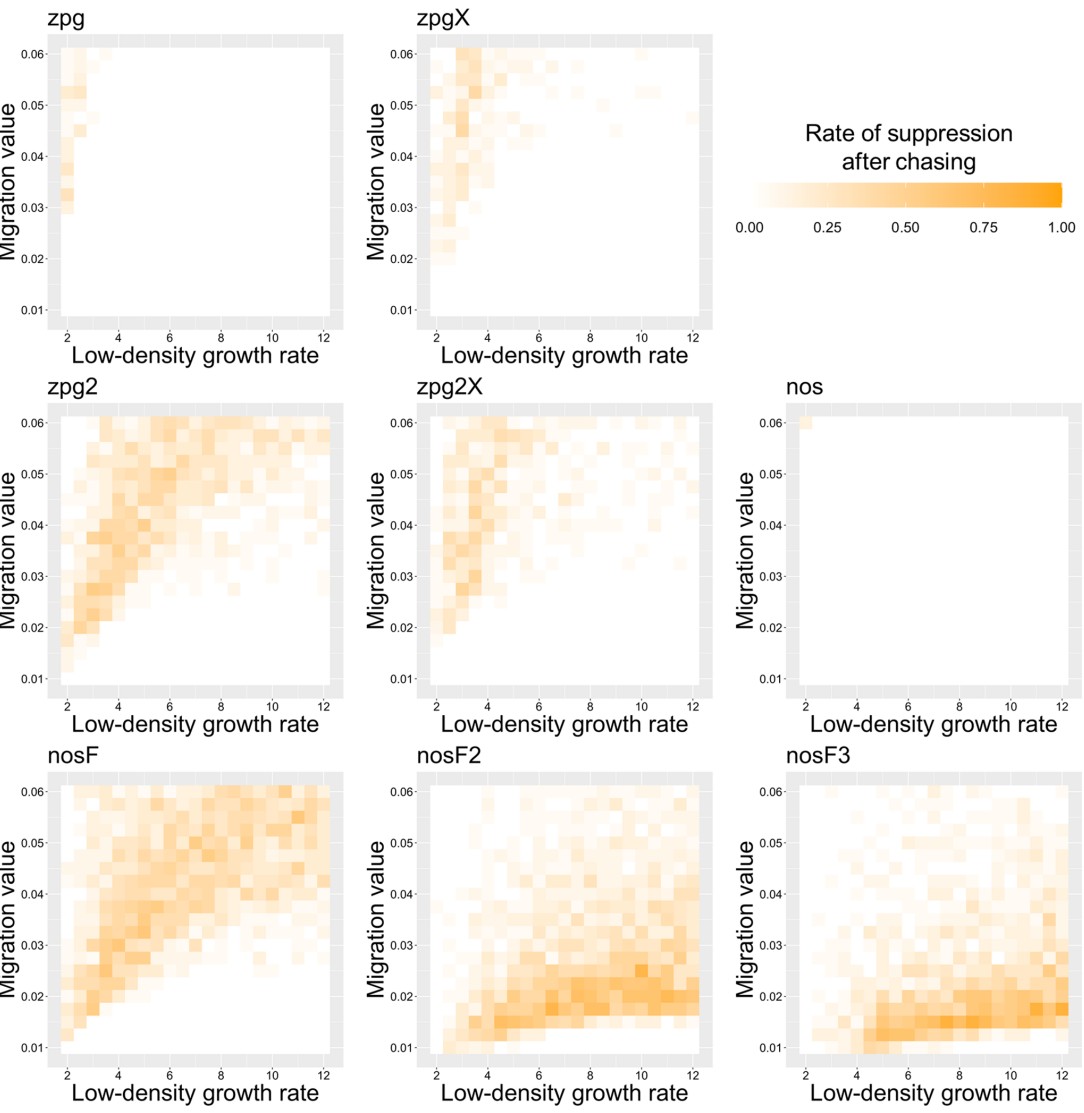

**Appendix 1—figure 4.** The rate of suppression after chasing in the discrete-generation model. Drive-carrying individuals were released into the middle of a wild-type population. The proportion of simulations in which suppression occurred after a chase that lasted a minimum of 10 generations is shown. Each point represents the average of 20 simulations.

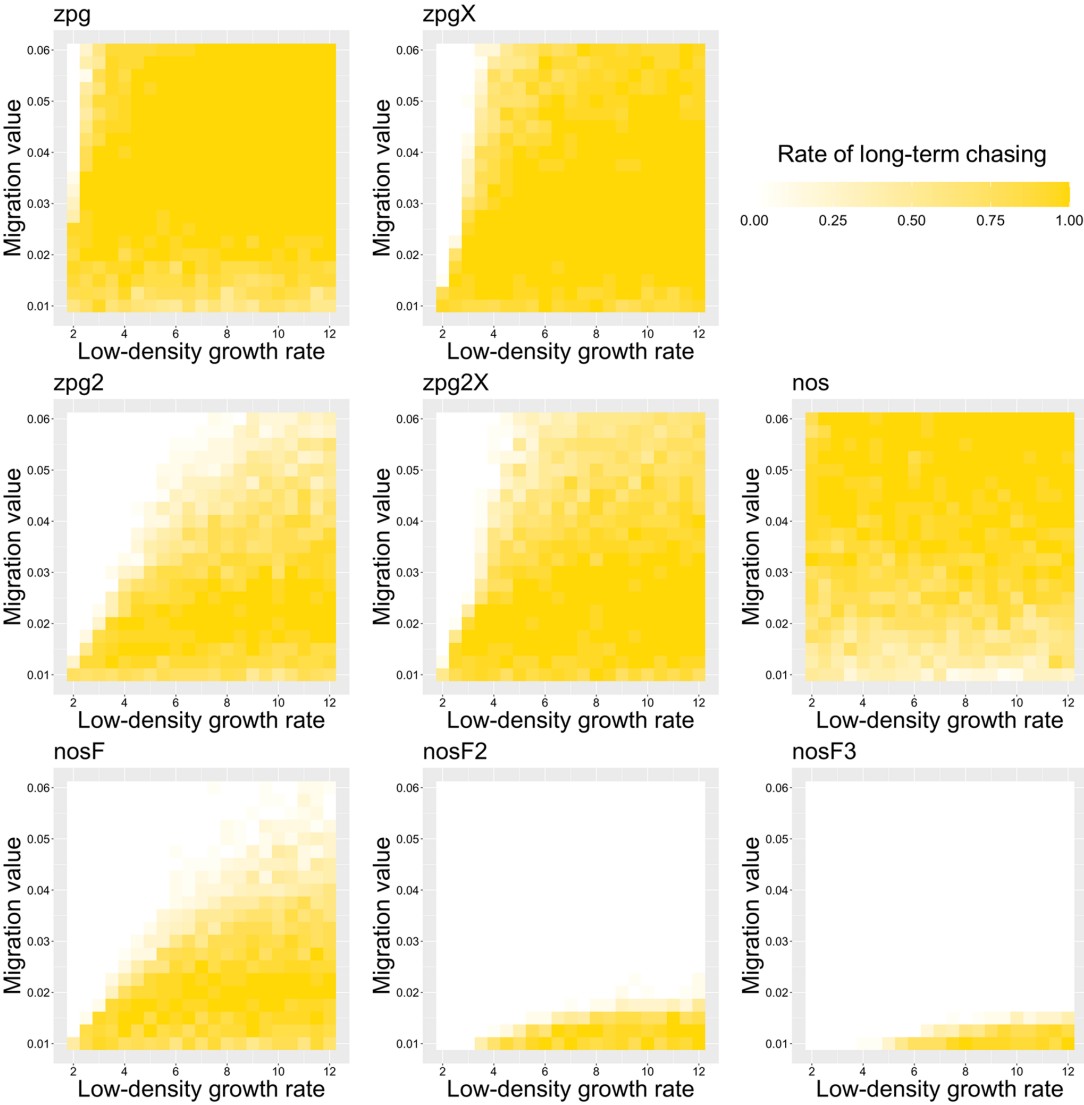

**Appendix 1—figure 5.** The rate of long-term chasing in the discrete-generation model. Drive-carrying individuals were released into the middle of a wild-type population. The proportion of simulations in which a long-term chasing outcome (defined by a chase continuing for 1000 generations after drive release) occurred is shown. Each point represents the average of 20 simulations.

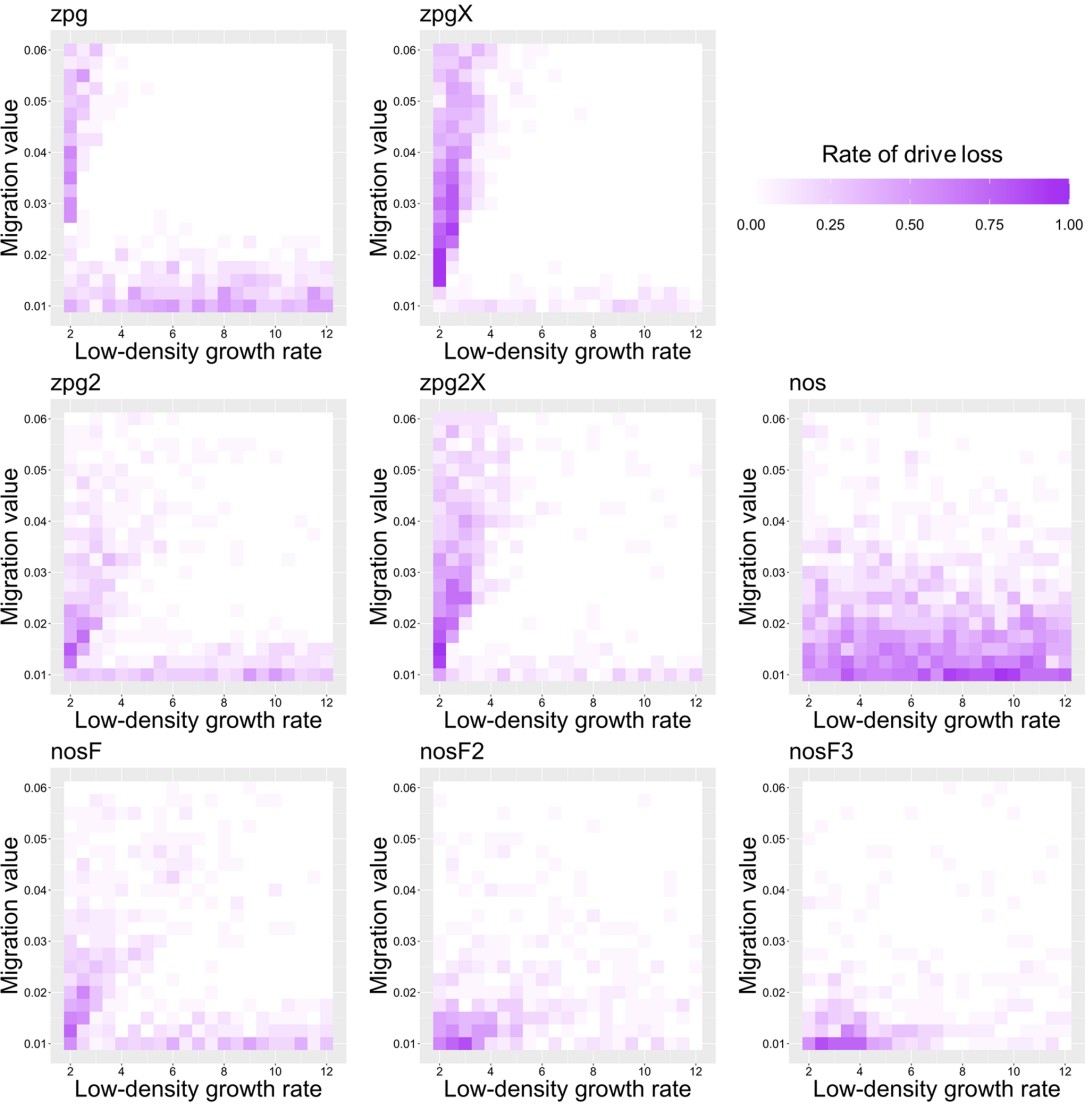

**Appendix 1—figure 6.** The rate of drive loss in the discrete-generation model. Drive-carrying individuals were released into the middle of a wild-type population. The proportion of simulations in which the drive was lost from the population is shown. Each point represents the average of 20 simulations.

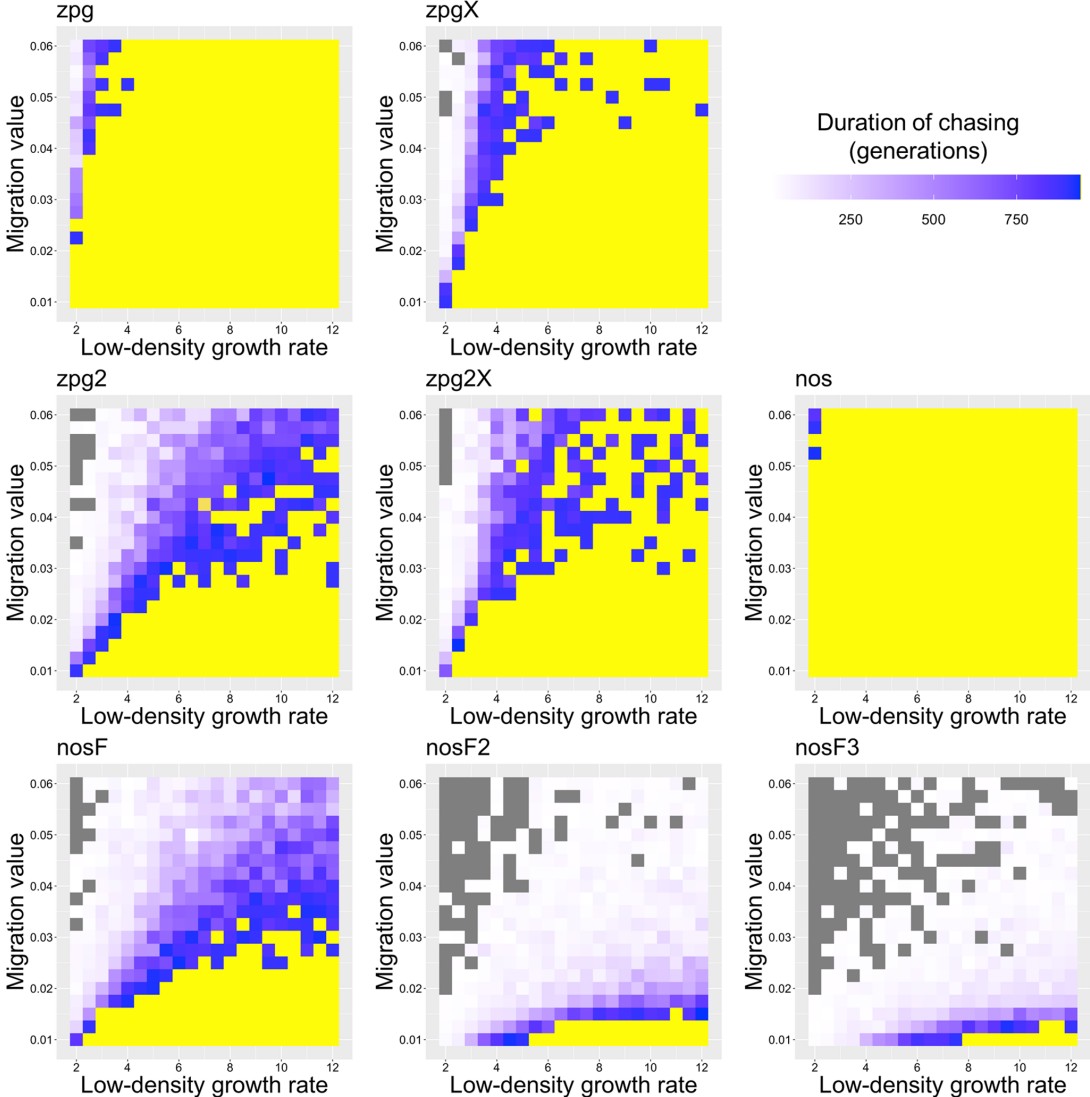

**Appendix 1—figure 7.** The duration of chasing prior to suppression in the discrete-generation model. Drive-carrying individuals were released into the middle of a wild-type population. The number of generations between the start of chasing and population elimination is shown. Each point represents the average of 20 simulations. Grey represents parameter combinations in which chasing did not occur in any simulation, and yellow represents parameter combinations in which chasing occurred but did not end in suppression in any simulation.

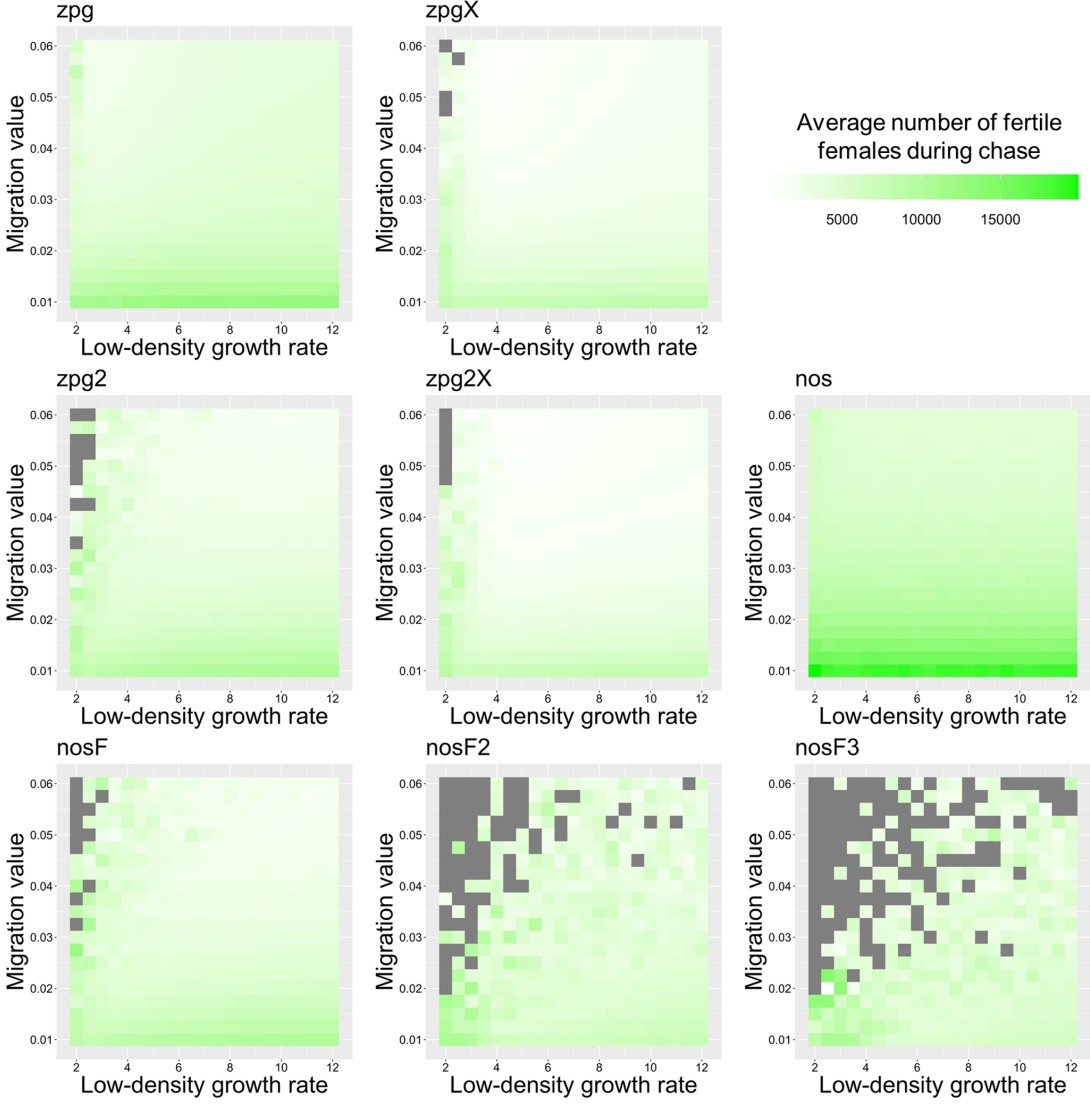

**Appendix 1—figure 8.** The average number of fertile females during chasing in the discrete-generation model. Drive-carrying individuals were released into the middle of a wild-type population consisting of an average of 25,000 females when at equilibrium. The average number of fertile females during periods of chasing is shown. Each point represents the average of 20 simulations. Grey represents parameter combinations in which chasing did not occur in any simulation.

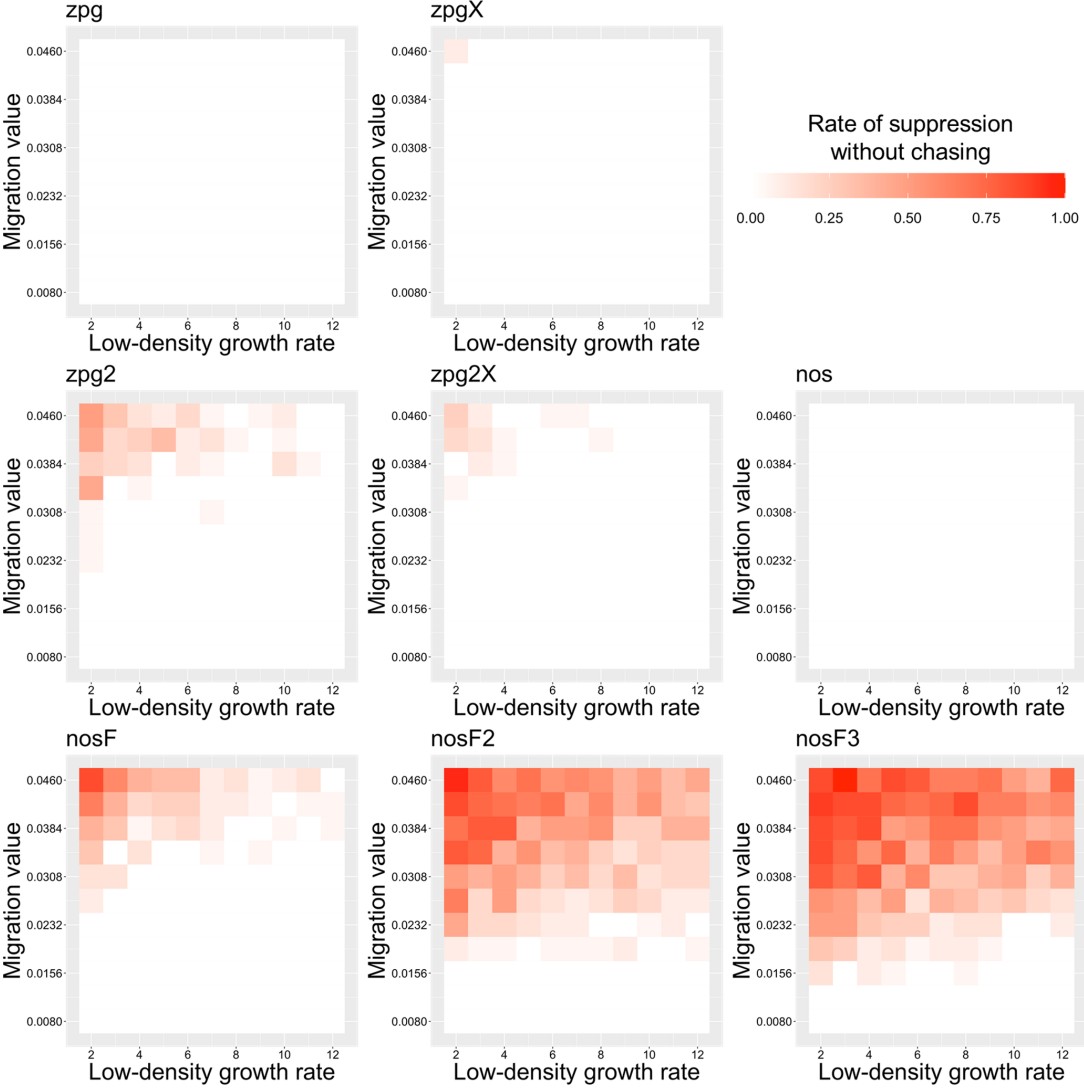

**Appendix 1—figure 9.** The rate of suppression without chasing in the *Anopheles* model. Drive-carrying mosquitoes were released into the middle of a wild-type population. The proportion of simulations in which suppression occurred either before a chase or within 32 weeks of the start of chasing is shown. Each point represents the average of 20 simulations.

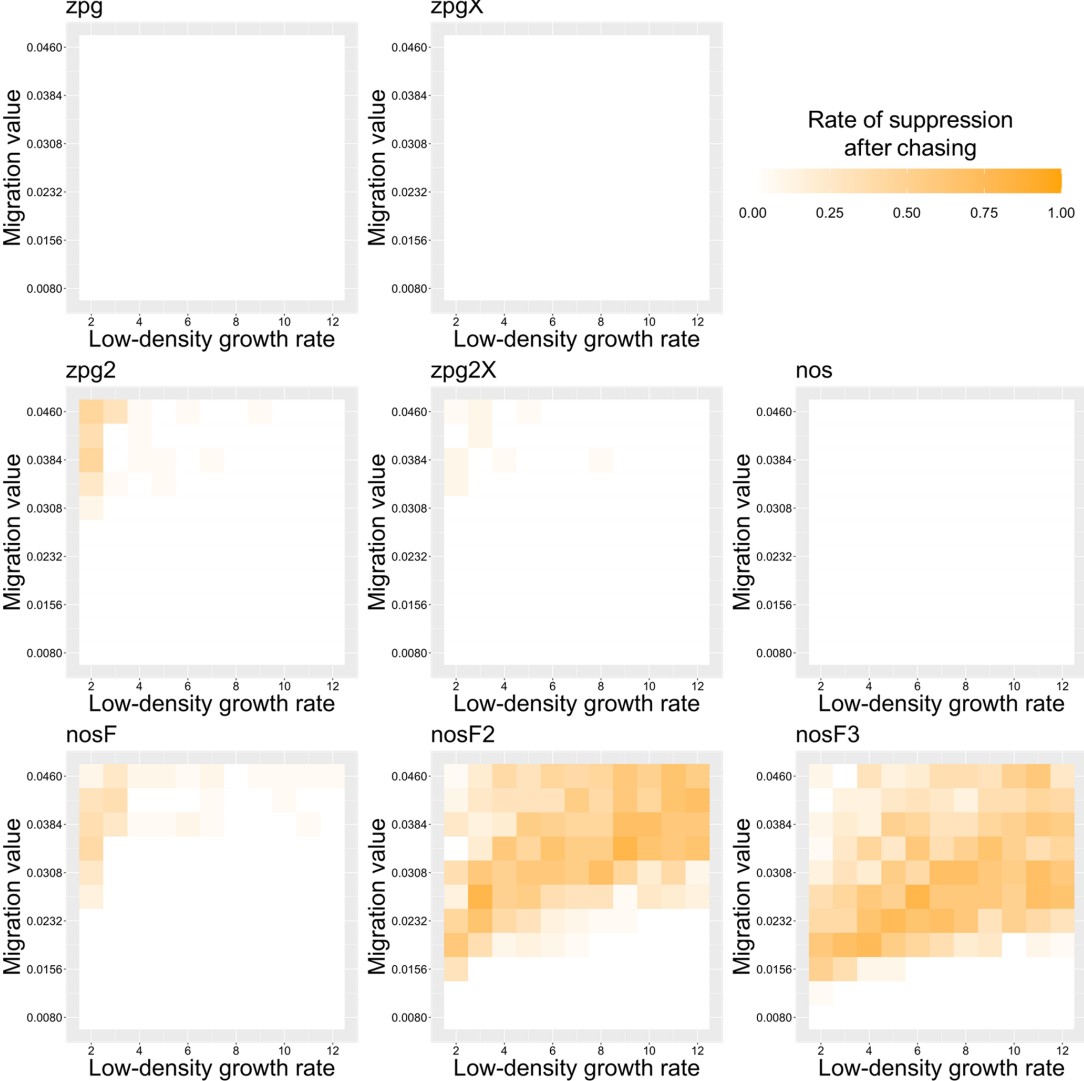

**Appendix 1—figure 10.** The rate of suppression after chasing in the *Anopheles* model. Drive-carrying mosquitoes were released into the middle of a wild-type population. The proportion of simulations in which suppression occurred after a chase that lasted a minimum of 32 weeks is shown. Each point represents the average of 20 simulations.

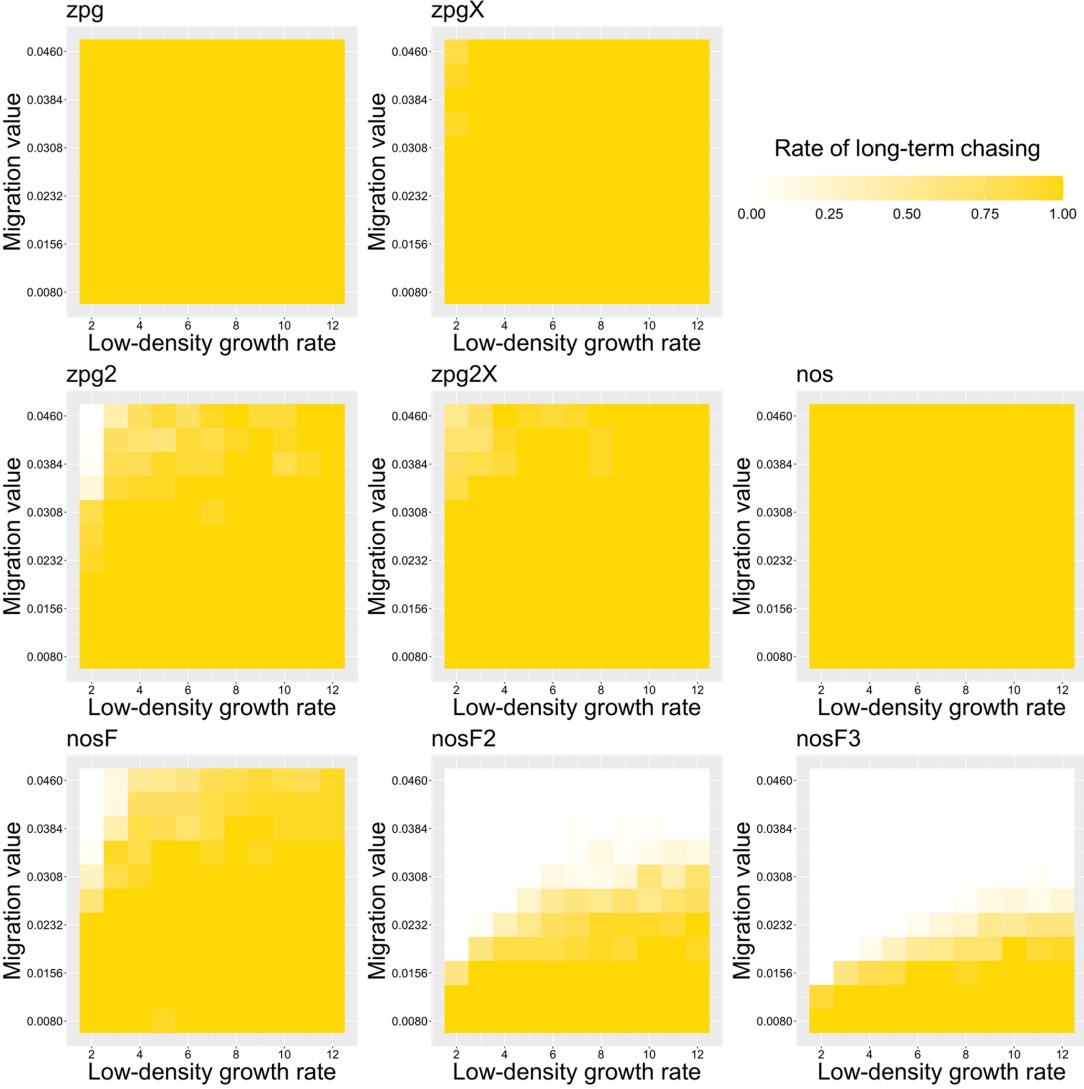

**Appendix 1—figure 11.** The rate of long-term chasing in the *Anopheles* model. Drive-carrying mosquitoes were released into the middle of a wild-type population. The proportion of simulations in which a long-term chasing outcome (defined by a chase continuing for 3167 weeks after drive release) occurred is shown. Each point represents the average of 20 simulations.

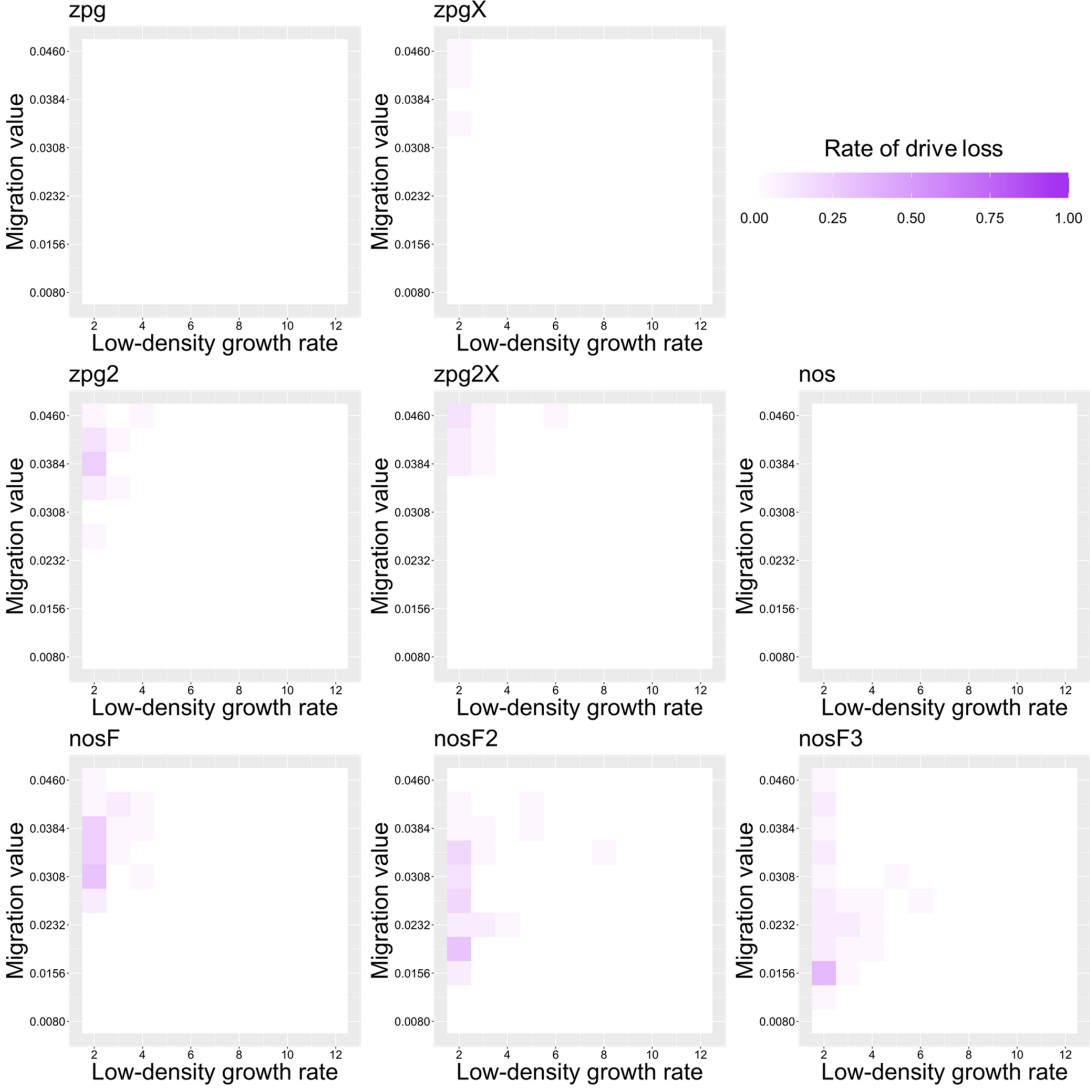

**Appendix 1—figure 12.** The rate of drive loss in the *Anopheles* model. Drive-carrying mosquitoes were released into the middle of a wild-type population. The proportion of simulations in which the drive was lost from the population is shown. Each point represents the average of 20 simulations.

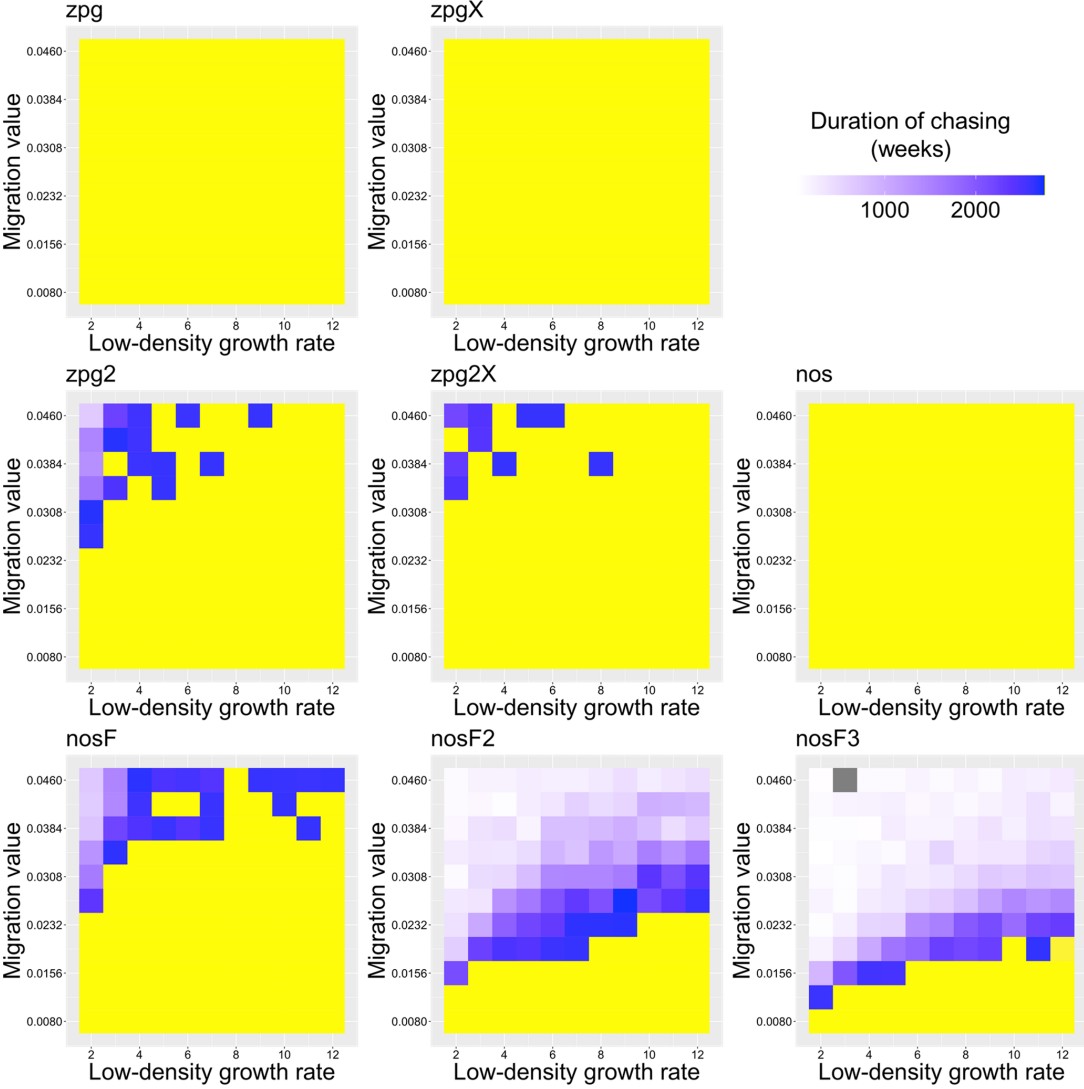

**Appendix 1—figure 13.** The duration of chasing prior to suppression in the *Anopheles* model. Drive-carrying mosquitoes were released into the middle of a wild-type population. The number of generations between the start of chasing and population elimination is shown. Each point represents the average of 20 simulations. Grey represents parameter combinations in which chasing did not occur in any simulation, and yellow represents parameter combinations in which chasing occurred but did not end in suppression in any simulation.

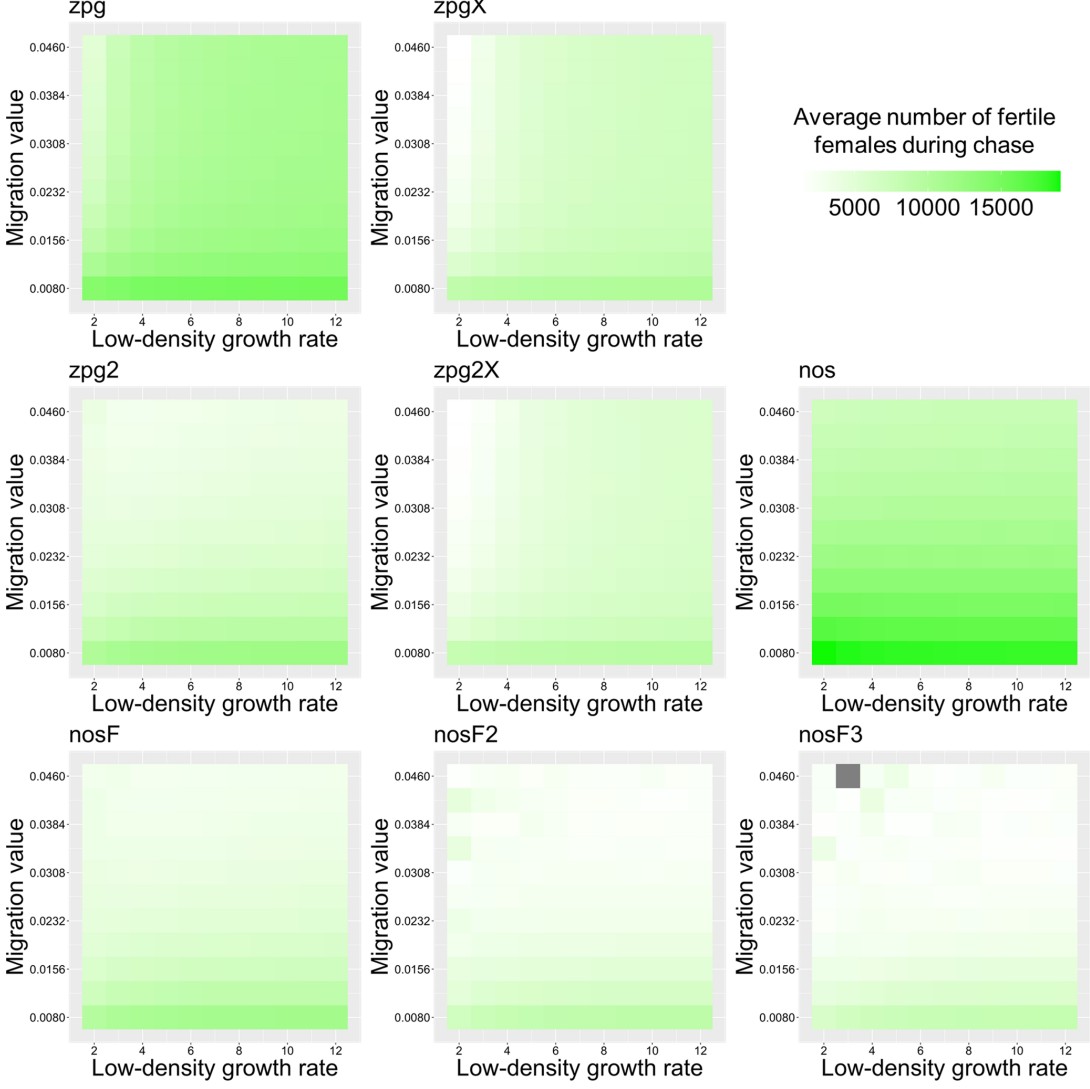

**Appendix 1—figure 14.** The average number of fertile females during chasing in the *Anopheles* model. Drive-carrying mosquitoes were released into the middle of a wild-type population consisting of an average of 25,000 females when at equilibrium. The average number of fertile females during periods of chasing is shown. Each point represents the average of 20 simulations. Grey represents parameter combinations in which chasing did not occur in any simulation.

