## [Editor Report]

This is a thorough, fundamental study assessing suppression gene drives against mosquitos. The models specifically consider the spatial dynamics of gene drives and whether a form of group selection may prevent the drive from eradicating the population, with mosquito ecology parameters, leading to compelling results. This manuscript will be of interest to those working in the technical development of gene drives, those predicting how such genetically modified insects would spread in the wild, and those evaluating the technology from regulatory and funding standpoints.

---

## [Decision Letter]

**Decision letter after peer review:**

Thank you for submitting your article "Finding the strongest gene drive: Simulations reveal unexpected performance differences between *Anopheles* homing suppression drive candidates" for consideration by *eLife*. Your article has been reviewed by 3 peer reviewers, and the evaluation has been overseen by a Reviewing Editor and George Perry as the Senior Editor. The following individuals involved in the review of your submission have agreed to reveal their identity: Sebald A.N.R Verkuijl (Reviewer #1); Jim Bull (Reviewer #3).

Essential revisions:

The reviewers are positive overall about your paper. As you will note from the full reviews further below, consistent comments included that model parameter value changes could lead to different quantitative results and the desire for a more explicit discussion of what is leading to the results. In our consultation discussion, there was a clear consensus that these and other points could be addressed with thoughtful text revision rather than major new analyses. In summary:

1. Expand the justification of your chosen parameters.

2. Expand discussion of the limitations of your study. Consider both of these points in the context of the perspective of the developers of the zpg drive.

3. Add a summary of what fundamental drive biology differences may be leading to your results in terms of the drive efficacy differences between the Anopheles-specific and general model. And is the seemingly detrimental effect of adding an X-shredder due to suppressing female abundances or to something else?

4. Given that gene drives are being evaluated for potential release suitability, the specific focus of this paper may cause it to attract considerable interest from a non-domain expert audience. As such, consider how you might make the writing more accessible to such an audience (including but not limited to explicitly discussing potential limitations of the study as mentioned above, and maybe even modeling in general).

5. Please also consider each individual item noted in the below reviews, and include how you addressed each comment in your point-by-point responses.

*Reviewer #1 (Recommendations for the authors):*

Sharing the model code is appreciated. I am unfamiliar with PhP and SLIM and was not able to evaluate the model code in the time I had (a limitation of my review, not a reflection of the paper/model).

The authors have gone through considerable effort to get parameter estimates from experimental data. This is very much appreciated. Table 1 gives an overview of the parameters used in the model and may give the impression that the distinction between the nos and zpg drives is substantial.

However, if it is reasonable to explore interpretations of drive data with a 0.45 difference in fitness, I don't think it is meaningful to make a distinction between the common core parameters of the nos and zpg drives (0.02 difference in male HDR, 0.01 difference in male germline resistance, 0.06 difference in maternal embryo resistance rates). It suggests those parameters are known to a far higher certainty than the parameters you vary from drive to drive.

In my opinion, this difference in core parameters needlessly obscures the comparisons between the nos and zpg drives. For example, I would suggest that zpg2 and nosF are practically equivalent conditions with regard to the accuracy of the parameter estimates and the limitations of any computational model. When there is a substantial difference between these conditions it says more about the sensitivity of the model outputs to the starting conditions than any meaningfully accurate estimate of how nos and zpg would differ under those conditions.

Another example: I understand that based on a certain interpretation of the experimental data it may be more reasonable to assume the nos drive has no fitness costs than that the zpg drive has no fitness cost. But simulating the no fitness cost scenario only for nos needlessly obscures the fact that is a practically identical approximation (again, in regards to the accuracy of the parameters and limitations of modelling) of how the zpg drive would perform with no fitness costs.

The authors may consider using the same core parameters for both drives making it easier for the reader to understand what truly underlies the differences between conditions. Allowing comparisons between all 8 conditions, instead of only within 2 groups of 4. Alternatively, the authors can make clear in the text that the difference between the zpg, and nos drives will mostly be due to the fitness/deposition/x-shredding parameters. That realisation also made it much easier for me to understand the results.

The above point may be superseded by this one: The parameters for the nos drive are taken at the nudel locus. The estimates for zpg-Cas9 are taken at the *dsx* locus. This is despite the fact that the zpg-Cas9 was also tested at the nudel locus and I think performed better there than at the *dsx* locus (the benefit of the *dsx* locus being the reduced possibility of functional resistance alleles). As such, I don't think this is a fair comparison, as the nos drive may well show the same reduction in fitness if moved to the *dsx* locus. And it would not be viable at the nudel locus due to the possibility of functional resistance alleles. The fact that the nos promoter has not been tested at the *dsx* locus is mentioned, but I think it has not been justified why the more 'fair' comparison at the nudel locus has not been used. Even if both nos and zpg would not perform well at the nudel locus in any real-world test, this study draws conclusions about their relative strength. So it still seems that is a more relevant condition.

*Reviewer #2 (Recommendations for the authors):*

My major concern is the Anopheles-specific model, including

1) The robusticity of the parameter estimates informed by literature (e.g., what are their likely confidence intervals), and

2) The sensitivity of the model to the different parameters chosen. A well-designed table can address point #1, as noted in the Public Review. For point #2, essentially what I would like to know is what is driving the difference between the outcomes of the discrete generation model and Anopheles model (Figure 1)? By rerunning the simulation with intermediate parameters between the two models, can you tell what is predisposing the Anopheles model to long-term chasing? You note that "[t]hese differences between models were likely at least partially due to the high reproductive capacity of Anopheles mosquitoes," but can this be shown by rerunning the Anopheles model with lower reproductive capacity? Given the stark differences between the outcomes of the two models and the centrality of this difference to your study, I feel such an addition would be useful.

*Reviewer #3 (Recommendations for the authors):*

Assuming I did not miss it, the paper lacks overviews of WHY the different constructs give rise to different outcomes. I think the authors should consider providing some kind of heuristic explanation of the main differences. For example, the inclusion of X-shredding with female sterility seems to hurt drive success. Is that for an ecological reason (e.g., the drive's greater efficacy on a local scale provides stronger group selection)? Alternatively, it might be something about drive specifics that is responsible for the effects described.

Since drive properties may change from lab to lab, it is worth telling the reader whether the important effects observed here are due to properties that may change with subtle improvements in engineering, or are instead due to basic ecological properties that have little to do with construction nuances.

trivia:

page 2 of the manuscript: 'we still lack a complete understanding of the effects …' can be said no matter what. The claim could be modified to have some meaning.

'chaotic' has a specific meaning in analysis. Is that what is meant here, or is the word chosen just to mean irregular?

page 3: 'Since this target gene is haplosufficient, female drive heterozygotes are potentially fully fertile' I imagine the model assumes full fertility, not potential full fertility.

page 4: first full paragraph. More detail could be used here. And I am guessing that the point of the dual strategy is that the drive causes an increase in the shredder, but the paragraph seems to omit this basic point.

'In the zpg and zpgX drives (but not the zpg2 and zpg2X drives)' -- I didn't easily find a description of what those are.

page 5: I'd like a bit more detail about how the model operates. Maybe a figure or table?

page 11: 'the drive must induce a sufficiently high genetic load in order to overpower the growth of wild-type populations at low density' Overpowering the growth of wild-type POPULATIONS? This is the panmictic model, so I would expect there are only wild-type genotypes. If my comment does not make sense, it's because I don't know what is being said.

Figure 1 legend: 'Offspring were artificially generated from fertile individuals at high rates to prevent complete population suppression ' I'm guessing this means that the population would have disappeared if fecundity hadn't been massively boosted. There might be a more direct way to say it. But why not let the population go extinct?

Page 12, top paragraph. Explain what (a) – (d) mean in population terms.

Page 15: 'The genetic load values measured by both models was within 1% for all of the drives (Figure 1)' Not clear -- what does 1% for all of the drives mean? The reference point is not clear.

---

## [Author Response]

Essential revisions:The reviewers are positive overall about your paper. As you will note from the full reviews further below, consistent comments included that model parameter value changes could lead to different quantitative results and the desire for a more explicit discussion of what is leading to the results. In our consultation discussion, there was a clear consensus that these and other points could be addressed with thoughtful text revision rather than major new analyses. In summary:1. Expand the justification of your chosen parameters.2. Expand discussion of the limitations of your study. Consider both of these points in the context of the perspective of the developers of the zpg drive.

We have added a discussion on the limitations of our input data in both the results and supplemental section. We have in particular expanded discussion of the error involved in our drive parameterization in a few locations in the manuscript (see responses to reviewer 1). We also reorganized our methods section and included more material to help readers understand the parameterization of our ecological data (much of this is still in the methods text, but the new Table S1 should assist with orienting readers).

3. Add a summary of what fundamental drive biology differences may be leading to your results in terms of the drive efficacy differences between the Anopheles-specific and general model. And is the seemingly detrimental effect of adding an X-shredder due to suppressing female abundances or to something else?

We have substantially expanded our Discussion section covering these points, including increased detail in our discussion of the X-shredder, an entirely new paragraph comparing our other drives, and increased detail in our paragraph considering differences between our discrete-generation and our *Anopheles* models.

4. Given that gene drives are being evaluated for potential release suitability, the specific focus of this paper may cause it to attract considerable interest from a non-domain expert audience. As such, consider how you might make the writing more accessible to such an audience (including but not limited to explicitly discussing potential limitations of the study as mentioned above, and maybe even modeling in general).

We have broken off the discussion of limitations into its own paragraph, giving it more prominence, as well as expanding it. We now conclude it with a new general discussion about the utility of models.

5. Please also consider each individual item noted in the below reviews, and include how you addressed each comment in your point-by-point responses.

Please see below for our individual responses.

Reviewer #1 (Recommendations for the authors):Sharing the model code is appreciated. I am unfamiliar with PhP and SLIM and was not able to evaluate the model code in the time I had (a limitation of my review, not a reflection of the paper/model).The authors have gone through considerable effort to get parameter estimates from experimental data. This is very much appreciated. Table 1 gives an overview of the parameters used in the model and may give the impression that the distinction between the nos and zpg drives is substantial.However, if it is reasonable to explore interpretations of drive data with a 0.45 difference in fitness, I don't think it is meaningful to make a distinction between the common core parameters of the nos and zpg drives (0.02 difference in male HDR, 0.01 difference in male germline resistance, 0.06 difference in maternal embryo resistance rates). It suggests those parameters are known to a far higher certainty than the parameters you vary from drive to drive.

In this study, our motivation was strongly derived from experimental findings regarding these specific drive types. This data is subject to uncertainty, and we agree that in many cases a 1-2% difference in measurements may in fact not be a difference at all. Yet, we felt that it would still be best for our fixed model parameters to match the experimental data as closely as possible.

While a precise uncertainty analysis of the experimental data is beyond the scope of this paper, we agree that we should call to attention this aspect of our input data. We have added this to our parameterization section. We now also state in the discussion that “parameterization of the drives was limited by our input data, with low sample sizes in particular causing high error in our estimates of nonfunctional resistance allele formation rates.” and that, “further experimental work could also reduce uncertainty regarding these drives, even after the basic mechanisms are resolved.” Finally, we now also tie this aspect more closely to the last part of our discussion in which we call for more experimental work to reduce measurement uncertainty due to the fact that small differences can produce large changes in ultimate outcomes. We have modified one sentence to read as follows, “Small parameter differences in drive performance could therefore be critical in ensuring drive success, suggesting that currently ambiguous drive characteristics should be thoroughly considered, and all drive parameters should be measured with as much accuracy as possible.”

In my opinion, this difference in core parameters needlessly obscures the comparisons between the nos and zpg drives. For example, I would suggest that zpg2 and nosF are practically equivalent conditions with regard to the accuracy of the parameter estimates and the limitations of any computational model. When there is a substantial difference between these conditions it says more about the sensitivity of the model outputs to the starting conditions than any meaningfully accurate estimate of how nos and zpg would differ under those conditions.Another example: I understand that based on a certain interpretation of the experimental data it may be more reasonable to assume the nos drive has no fitness costs than that the zpg drive has no fitness cost. But simulating the no fitness cost scenario only for nos needlessly obscures the fact that is a practically identical approximation (again, in regards to the accuracy of the parameters and limitations of modelling) of how the zpg drive would perform with no fitness costs.

We structured the paper around these particular drive constructs with “core parameters” because these sets of parameters represent our best estimates for how the drives behave. Where qualitative differences were present, we fixed as many parameters as possible and varied the remaining parameters in order to provide alternative explanations that might be similarly good representations for how the drives have been measured to behave. It is for this reason that male deposition and somatic fitness costs are varied in the *zpg* drives but not the *nos* drives, and that somatic fitness cost in both sexes is varied in the *nos* drives but not the *zpg* drives. The eight resultant variants are thus not just intended as eight well distributed points from a parameter space, but as eight actually possible drives that could be selected from, pending a more complete determination of which parameterizations are most accurate

If the focus of this manuscript were a full exploration of the parameter space, we would certainly simulate all of the combinations mentioned in this comment, likely producing more heatmaps to allow more parameters to vary. Yet, we believe a core appeal of this project is in relating back to the specific set of drive constructs that have actually been engineered or could be engineered by combining elements that have actually been engineered. In a previous study, we put more effort into continuously varying drive performance parameters (Champer, Kim, et al., in Molecular Ecology, 2021).

That said, we agree that readers should be directed toward considering these points. In addition to the above listed changes, we now include a full paragraph at the end of our supplemental parameterization set that reads:

“We note that all performance parameters for the nosF drive are very similar to the zpg2 drive parameterization. While sample sizes for drive conversion and for fitness calculations are fairly high, our estimate of germline resistance allele formation is based on common modeling convention rather than robust experimental evidence, and our estimate for the embryo cut rate are based on modest sample sizes. Thus, the exact differences between these drives are somewhat uncertain and would require additional experimentation to confirm. Other differences between our drives’ parameter sets are based on qualitative assumptions that provide a clearer contrast for interpretation of results.”

The authors may consider using the same core parameters for both drives making it easier for the reader to understand what truly underlies the differences between conditions. Allowing comparisons between all 8 conditions, instead of only within 2 groups of 4. Alternatively, the authors can make clear in the text that the difference between the zpg, and nos drives will mostly be due to the fitness/deposition/x-shredding parameters. That realisation also made it much easier for me to understand the results.

Another way to think about this is that are eight are not exclusively meant to be compared together. They are presented together, but this presentation is the foreground to a set of separate stories. One of these stories is that of determining which interpretation of the zpg drive is most realistic (male deposition versus somatic fitness costs). Another is whether supplementation with an X-shredder is helpful (admittedly, this does depend somewhat on zpg mechanism, so even though we find the male deposition interpretation less plausible, we still combine it with the Xshredder to discuss how this would change things). This thread of our work ends with us favoring the interpretation of zpg with somatic fitness as the more, realistic and determining that a drive without the X-shredder would have superior performance.

In parallel, we reject male somatic fitness for nos and identify two reasonable parameter sets. One of these is highly similar to zpg (perhaps within the bounds of uncertainty of the parameters) as the reviewer suggests (the drive conversion difference may be a little more reliable, but the germline resistance is just an estimate based on common modeling convention, and the embryo resistance estimate is based on low sample sizes). This difference is still worth some discussion because it shows that some parameters seem to be more important than others. Another element of this goal of the paper is the comparison to the interpretation of the nos drive with reduced female somatic fitness cost, which helps the drive quite a bit. The final nos drive without fitness costs is within the realm of possibility and extends the investigation of somatic fitness cost, but it is also partially included simply because we also wish to show how a drive with near-optimum characteristics can perform!

A restructuring of the modeling to use the same core parameters might slightly reduce the complexity and total amount of data, but it would also remove a potentially interesting zpg vs nos comparison and perhaps move us slightly further away from a paradigm in which we are attempting to strongly base our paper on experimental results. We believe our choice to base the paper on experimental results will be of greater general interest.

The above point may be superseded by this one: The parameters for the nos drive are taken at the nudel locus. The estimates for zpg-Cas9 are taken at the dsx locus. This is despite the fact that the zpg-Cas9 was also tested at the nudel locus and I think performed better there than at the dsx locus (the benefit of the dsx locus being the reduced possibility of functional resistance alleles). As such, I don't think this is a fair comparison, as the nos drive may well show the same reduction in fitness if moved to the dsx locus. And it would not be viable at the nudel locus due to the possibility of functional resistance alleles. The fact that the nos promoter has not been tested at the dsx locus is mentioned, but I think it has not been justified why the more 'fair' comparison at the nudel locus has not been used. Even if both nos and zpg would not perform well at the nudel locus in any real-world test, this study draws conclusions about their relative strength. So it still seems that is a more relevant condition.

Many performance parameters are largely functions of the Cas9 promoter, though target sequence, target gene, and genomic location can certainly be quite important too. We now make it explicit in our main text that *nos* is parameterized based on a different target gene. In fact, at this other target gene, the *zpg* drives performed worse than at *dsx*, so it is possible that *nos* would also actually perform better if designed to target *dsx*.

We have expanded our discussion of this in the supplemental section about drive parameterization. It now reads:

“The *nos* promoter represents a potential alternative to *zpg*’s germline-restricted expression of Cas9. We parameterize this drive based on a previous study^18^ that involved targeting the *nudel* gene. Thus, a *nos*-based drive at *dsx* may actually have somewhat different performance parameters (*zpg* based drives were slightly more effective at *dsx*^6^ than at *nudel*^18^).”

Reviewer #2 (Recommendations for the authors):My major concern is the Anopheles-specific model, including1) The robusticity of the parameter estimates informed by literature (e.g., what are their likely confidence intervals), and2) The sensitivity of the model to the different parameters chosen. A well-designed table can address point #1, as noted in the Public Review.

For some ecological parameters, confidence intervals don’t exist. Even for our drive parameters, we often could not access the raw data we would need to estimate confidence intervals. However, we have added additional discussion of certain drive performance estimates (see responses to reviewer 1 and accompanying manuscript text changes). We have also added Table S1 to provide an overview of our model parameters.

For sensitivity of some ecological parameters, we did conduct an analysis varying migration rates and low-density growth rates. For some drive parameters, these can be found in our earlier study using the discrete-generation model (Champer, Kim, et al., in Molecular Ecology, 2021 – the exact numbers would be different in the mosquito model, but the parameter effects themselves would likely be similar).

For point #2, essentially what I would like to know is what is driving the difference between the outcomes of the discrete generation model and Anopheles model (Figure 1)? By rerunning the simulation with intermediate parameters between the two models, can you tell what is predisposing the Anopheles model to long-term chasing? You note that "[t]hese differences between models were likely at least partially due to the high reproductive capacity of Anopheles mosquitoes," but can this be shown by rerunning the Anopheles model with lower reproductive capacity? Given the stark differences between the outcomes of the two models and the centrality of this difference to your study, I feel such an addition would be useful.

There are three major differences between these models. One of them is the overlapping generations in the *Anopheles* model. Another is inclusion of a lifecycle in the *Anopheles* model. We suspect that these two differences are not likely to have a large effect on simulation outcomes. The third difference is in the details of density-dependent reproduction. In the discrete-generation model, density affects female fecundity. In the *Anopheles* model, density affects larval viability. We believe that this likely accounts for the main differences between the model outcomes and was what we were alluding to when discussing the high reproductive capacity of mosquitoes (the fundamental reproductive capacity as a function of density is actually similar between the models with the same density curve). In retrospect, this was not the clearest way to explain this, and we have adjusted the text, which now reads:

“These differences between models were likely at least partially due to the fact that competition in the *Anopheles* model affects offspring viability rather than female fecundity. Thus, fertile *Anopheles* individuals are not inhibited in reproduction by sterile individuals like fertile females are in the discrete-generation model. This ensures a more robust population even under pressure from a drive with a high genetic load when many sterile or otherwise non-contributing individuals (such as excess males due to an X-shredder) are present.”

We have also added (later in the same paragraph):

“Finally, the higher number of larvae generated by identical adult populations in the *Anopheles* model can reduce the chance of stochastic elimination compared to the discrete-generation model, even in panmictic populations^19^.”

We hope this gives some additional flavor of a possible explanation for readers interested in the “why” behind differences in model outcomes.

As for really getting to the bottom of this, we could potentially relax combinations of these three differences between the models, but this would be very computationally intensive and likely requires a sophisticated analysis. It would certainly be quite interesting and likely a worthy topic for a future “fundamentals” manuscript about spatial suppression outcomes. However, in the current manuscript, we are more focused on skipping straight to a continuous space model that is our best attempt at a realistic mosquito population in which we can analyze our specific drive candidates. We have added the following text to the discussion:

“Future studies could more precisely assess how different fundamental design decisions in continuous space models (e.g., the type of spatial competition and what stages of life competition occurs at) can impact the predicted outcome of different types of suppression drive releases.”

Reviewer #3 (Recommendations for the authors):Assuming I did not miss it, the paper lacks overviews of WHY the different constructs give rise to different outcomes. I think the authors should consider providing some kind of heuristic explanation of the main differences. For example, the inclusion of X-shredding with female sterility seems to hurt drive success. Is that for an ecological reason (e.g., the drive's greater efficacy on a local scale provides stronger group selection)? Alternatively, it might be something about drive specifics that is responsible for the effects described.Since drive properties may change from lab to lab, it is worth telling the reader whether the important effects observed here are due to properties that may change with subtle improvements in engineering, or are instead due to basic ecological properties that have little to do with construction nuances.

This is a complicated question, with potentially different explanations for each comparison. We discussed several aspects of these, but not necessarily in a central location in the manuscript (some of it was scattered in the results). We want to emphasize that for basic comparisons between fundamentally different types of drives, we cover this in much more detail in Champer, Kim, et al., in Molecular Ecology 2021. For this study, there are three important points that we believe account for performance differences:

1. Genetic load.

2. Presence of an X-shredder.

3. Inefficiency from somatic fitness costs.

We have expanded our X-shredder paragraph for increased clarity. We have also added another completely new paragraph to the discussion that further discusses differences between drives in a more focused fashion.

trivia:page 2 of the manuscript: 'we still lack a complete understanding of the effects …' can be said no matter what. The claim could be modified to have some meaning.

The text has been modified to be more specific.

'chaotic' has a specific meaning in analysis. Is that what is meant here, or is the word chosen just to mean irregular?

This was chosen to mean “unpredictable”, which describes patterns of chasing, which could turn out differently even with the same starting condition. We have changed the wording as per the reviewer’s advice to avoid confusion with the mathematic definition of chaotic.

page 3: 'Since this target gene is haplosufficient, female drive heterozygotes are potentially fully fertile' I imagine the model assumes full fertility, not potential full fertility.

The text has been modified for clarity. Somatic fitness effects can potentially reduce female fertility.

page 4: first full paragraph. More detail could be used here. And I am guessing that the point of the dual strategy is that the drive causes an increase in the shredder, but the paragraph seems to omit this basic point.

A longer explanation of the benefits of the combined system has been added (specifically, how the two systems help overcome each other’s weakness). The new text is as follows:

“Overall, the X-shredder reduces the number of drive females, reducing the influence of drivebased somatic fitness costs in females. Unlike a Driving Y, this autosomal X-shredder cannot increase its own inheritance. It relies on the linked homing drive element for this purpose.”

'In the zpg and zpgX drives (but not the zpg2 and zpg2X drives)' -- I didn't easily find a description of what those are.

We mistakenly referenced these drives before actually defining them (in the following paragraph). The text has now been modified.

page 5: I'd like a bit more detail about how the model operates. Maybe a figure or table?

We have added a figure (new Figure 1), which will hopefully clarify our drive performance parameters and mechanisms.

page 11: 'the drive must induce a sufficiently high genetic load in order to overpower the growth of wild-type populations at low density' Overpowering the growth of wild-type POPULATIONS? This is the panmictic model, so I would expect there are only wild-type genotypes. If my comment does not make sense, it's because I don't know what is being said.

We agree that reference to wild-type alleles was confusing. We have removed this, and the text now reads, “To eliminate a panmictic population, the drive must induce a sufficiently high genetic load in order to overpower the increased population growth rate at low density.”

Figure 1 legend: 'Offspring were artificially generated from fertile individuals at high rates to prevent complete population suppression ' I'm guessing this means that the population would have disappeared if fecundity hadn't been massively boosted. There might be a more direct way to say it. But why not let the population go extinct?

A detailed description of this method is present in the supplemental methods section of the manuscript, along with a bit more explanation of why we elected to use this method in lieu of just letting the population go extinct. We now refer to this supplemental section in Figure 1 and in Figure S1. In short, we need to measure the genetic load when the drive is at equilibrium, and it won’t reach equilibrium if the population is eliminated. The method we use to accomplish this should not affect average genetic load measurements or average frequency trajectories, only population sizes.

Page 12, top paragraph. Explain what (a) – (d) mean in population terms.

The discussion of these possible results has been expanded as per the reviewer’s suggestion.

Page 15: 'The genetic load values measured by both models was within 1% for all of the drives (Figure 1)' Not clear -- what does 1% for all of the drives mean? The reference point is not clear.

The text has been modified for greater clarity.